# Improving the Euclidean Diffusion Generation of Manifold Data by Mitigating Score Function Singularity

Zichen Liu [1], Wei Zhang [*,2], and Tiejun Li [*,1,3,4]

[1]Center for Data Science, Peking University
[2]Zuse Institute Berlin
[3]LMAM and School of Mathematical Sciences, Peking University
[4]Center for Machine Learning Research, Peking University

## Abstract

Euclidean diffusion models have achieved remarkable success in generative modeling across diverse domains, and they have been extended to manifold cases in recent advances. Instead of explicitly utilizing the structure of special manifolds as studied in previous works, in this paper we investigate direct sampling of the Euclidean diffusion models for general manifold-structured data. We reveal the multiscale singularity of the score function in the ambient space, which hinders the accuracy of diffusion-generated samples. We then present an elaborate theoretical analysis of the singularity structure of the score function by decomposing it along the tangential and normal directions of the manifold. To mitigate the singularity and improve the sampling accuracy, we propose two novel methods: (1) Niso-DM, which reduces the scale discrepancies in the score function by utilizing a non-isotropic noise, and (2) Tango-DM, which trains only the tangential component of the score function using a tangential-only loss function. Numerical experiments demonstrate that our methods achieve superior performance on distributions over various manifolds with complex geometries.

## 1  Introduction

Diffusion models [11, 35, 36, 38] have demonstrated remarkable success in generative modeling in various domains, including image generation [12], audio synthesis [5], molecule generation [13], and other applications [22, 48]. These models operate through a forward process, which gradually perturbs data into noise, and a reverse process, which reconstructs the data from noise.

Beyond Euclidean space, many scientific fields involve data distributions constrained on Riemannian manifolds. Examples include geographical sciences [28] using spheres, protein structures [45] and robotic movements [34] with $SE(3)$ and $SO(3)$, lattice quantum chromodynamics [26] with $SU(3)$, 3D computer graphics [14] with triangular meshes, and cell development [21] with the Poincaré disk. To model such data, recent works [8, 17, 26, 49] have extended diffusion models to Riemannian manifolds, achieving notable progress by leveraging manifold geometry. Nonetheless, these methods face challenges due to the computational complexity of simulating diffusion and obtaining accurate geometric information like the heat kernel.

On the other hand, a natural idea is to directly apply diffusion models designed for Euclidean space to manifold-structured data. However, this approach encounters a fundamental challenge due to the

---

[*]Corresponding author: tieli@pku.edu.cn (T. Li), wei.zhang@fu-berlin.de (W. Zhang)

39th Conference on Neural Information Processing Systems (NeurIPS 2025).

singularity of the score function. Prior studies [3, 27, 29] have identified that under the manifold hypothesis, the norm of the score function explodes as the time in the reverse process approaches zero. Recent works [4, 27, 29] have theoretically analyzed this singularity under the Ornstein-Uhlenbeck process, providing asymptotic bounds on the score function and introducing structural assumptions on the data distribution. In this paper, we further enhance these insights and propose methods for mitigating the singularity of the score function under the manifold hypothesis.

**Our work.** We consider a probability distribution on a known $d$-dimensional submanifold $\mathcal{M}$ embedded in $\mathbb{R}^n$, where $d < n$. Given samples from this distribution, our objective is to generate new samples by directly applying Euclidean diffusion models. Within the framework of Variance Exploding Stochastic Differential Equations (VESDE) with such a manifold setting, we theoretically demonstrate that the perturbed score function exhibits distinct scales along the tangential and normal components. To address this issue, we propose the following two methods to improve diffusion models (DM):

1. **Niso-DM**: Perturb data with non-isotropic noise by introducing additional noise along the normal direction during the forward diffusion process. This reduces the scale discrepancy in the score function between the tangential and normal directions, making the relaxed score function easier to approximate.

2. **Tango-DM**: Train only the tangential component of the score function using a tangential-only loss function. By bypassing the training of the normal component, this method overcomes the learning difficulties caused by the multiscale issue.

Our main contributions are as follows:

- We present an elaborate theoretical analysis of the multiscale singularity structure of the score function by separating it along the tangential and normal directions of the manifold in the embedded Euclidean space. Furthermore, we investigate the relationship between the full-space score function and the original Riemannian score function, which has not been considered before.

- We propose two novel methods, Niso-DM and Tango-DM, making the relaxed score function easier to approximate. We also give a theoretical analysis of Niso-DM. Empirically, our methods demonstrate superior performance on distributions across various complex and non-trivial manifolds. This is in sharp contrast with previous methods that can only handle manifolds with special analytical structure.

## 2 Backgrounds and preliminaries

### 2.1 Diffusion models

Diffusion models can be formulated using stochastic differential equations (SDEs) [38]. Specifically, the data perturbation process is modeled by the following forward SDE:

$$\mathrm{d}X_t = f(X_t, t)\mathrm{d}t + g(t)\mathrm{d}W_t, \tag{1}$$

where $f$ and $g$ are fixed drift and diffusion coefficients, respectively. By carefully selecting $f$ and $g$, the distribution of $X_T$, with the probability density function (pdf) $p_T$, can be well approximated by a Gaussian distribution $\mathcal{N}(0, \sigma_T^2 I)$, where $\sigma_T^2$ represents the variance of the Gaussian distribution. The corresponding reverse-time SDE, which transports $p_T$ back to the initial pdf $p_0$, is given by:

$$\mathrm{d}X_t = \left(f(X_t, t) - g(t)^2 \nabla_x \log p_t(X_t)\right)\mathrm{d}t + g(t)\mathrm{d}\overline{W}_t, \tag{2}$$

where $\overline{W}_t$ is a standard Brownian motion in reverse time and $p_t$ is the time-dependent distribution driven by the stochastic process $X_t$.

Inspired by this formulation, diffusion models employ a neural network $s_\theta(x, t)$ to approximate the score function $\nabla_x \log p_t(x)$. Since $p_t(x)$ is analytically unknown, direct optimization of the quadratic loss:

$$\mathcal{L}_{\mathrm{quad}}(\theta) = \mathbb{E}_{t \sim \mathcal{U}(0,1)}[\lambda_t \ell_{\mathrm{quad}}(t, \theta)], \tag{3}$$

$$\ell_{\mathrm{quad}}(t, \theta) = \mathbb{E}_{x_t} \|s_\theta(x_t, t) - \nabla_{x_t} \log p_t(x_t)\|^2, \tag{4}$$

is intractable. Instead, score matching techniques such as denoising score matching [43] are commonly used, which learn the score with the loss

$$\ell_{\mathrm{dsm}}(t, \theta) = \mathbb{E}_{x, x_t} \| s_\theta(x_t, t) - \nabla_{x_t} \log p_t(x_t|x) \|^2, \tag{5}$$

where $p_t(x_t|x)$ denotes the probability density of $X_t$ in (1) conditioned on the initial state $X_0 = x$. To utilize (5), the following two specific SDEs introduced in [38] are often considered, in which cases the density $p_t(x_t|x)$ has a closed-form.

1. Variance Preserving SDE (VPSDE):

$$\mathrm{d}X_t = -\frac{1}{2}\beta(t)X_t\mathrm{d}t + \sqrt{\beta(t)}\mathrm{d}W_t, \tag{6}$$

   where $\beta(t) > 0$.

2. Variance Exploding SDE (VESDE):

$$\mathrm{d}X_t = \sqrt{\frac{\mathrm{d}\sigma_t^2}{\mathrm{d}t}}\mathrm{d}W_t, \tag{7}$$

   where $\sigma_t$ is a predefined noise scale, commonly chosen as $\sigma_t = \sigma_{\min}(\sigma_{\max}/\sigma_{\min})^{t/T}$ for some $\sigma_{\max} > \sigma_{\min} > 0$. In this case, $p_t(x_t|x)$ follows a Gaussian distribution $\mathcal{N}(x_t|x, \sigma_t^2 I)$.

In this study, we choose VESDE for its advantageous properties. Under VESDE, the mean of the perturbed distribution remains unchanged. At any time, the perturbed distribution $p_t(x)$ can be viewed as points on the original manifold $\mathcal{M}$ with added noise. This simplifies theoretical analysis and avoids the complications introduced by the evolving manifold in VPSDE. In fact, VPSDE involves an evolving manifold over time, defined as $\mathcal{M}_t = \{e^{-\frac{1}{2}\int_0^t \beta(s)\mathrm{d}s}x \mid x \in \mathcal{M}\}$ as the mean with noise perturbation.

## 2.2 Riemannian manifolds and notation

In this paper, the manifold $\mathcal{M}$ is viewed as $\mathcal{M} = \{x \in \mathbb{R}^n | \xi(x) = 0\}$ embedded in $\mathbb{R}^n$, $\xi : \mathbb{R}^n \to \mathbb{R}^{n-d}$ is a known function, and we assume that $\nabla\xi(x) \in \mathbb{R}^{n\times(n-d)}$ has full column rank for all $x \in \mathcal{M}$. The target distribution is formulated as $p_0(x)\mathrm{d}\sigma_{\mathcal{M}}(x)$, where $p_0(x)$ is an unknown density and $\mathrm{d}\sigma_{\mathcal{M}}(x)$ represents the volume form [32] of $\mathcal{M}$. The unit normal vectors at a point $x \in \mathcal{M}$ are given by $N(x) = \nabla\xi(x)\left(\nabla\xi(x)^T\nabla\xi(x)\right)^{-\frac{1}{2}}$. For $x \in \mathcal{M}$, the operator $\nabla_x^{\mathcal{M}}$ is defined as $\nabla_x^{\mathcal{M}}h(x) = P(x)\nabla_x h(x)$ for any smooth function $h$, where $P(x)$ is the projection matrix [32] onto the tangent space of $\mathcal{M}$ and is given by

$$P(x) = I - N(x)N(x)^T = I - \nabla\xi(x)(\nabla\xi(x)^T\nabla\xi(x))^{-1}\nabla\xi(x)^T. \tag{8}$$

## 3 Scale discrepancy of score functions

It is commonly understood that singularities arise when Euclidean diffusion models are directly applied to data with manifold structures. In this section, we examine the singularity of diffusion models under VESDE, following the settings in Section 2.2. By introducing Gaussian noise across the entire space, the perturbed data become distributed throughout $\mathbb{R}^n$, deviating from their original confinement to the $d$-dimensional submanifold $\mathcal{M}$. The perturbed distribution is given by

$$p_\sigma(\tilde{x}) = \int_{\mathcal{M}} p_0(x)p_\sigma(\tilde{x}|x)\mathrm{d}\sigma_{\mathcal{M}}(x) = (2\pi\sigma^2)^{-\frac{n}{2}}\int_{\mathcal{M}} p_0(x)e^{-\frac{|x-\tilde{x}|^2}{2\sigma^2}}\mathrm{d}\sigma_{\mathcal{M}}(x), \tag{9}$$

where $\tilde{x}$ is a variable in the embedded space $\mathbb{R}^n$, $p_\sigma(\tilde{x}|x)$ denotes the pdf $\mathcal{N}(x, \sigma^2 I)$ of the isotropic perturbtion, and $p_0(x)$ is only defined on $\mathcal{M}$. Hereafter, the subscript of $p$ denotes the noise scale $\sigma$ instead of the time $t$.

When the noise scale $\sigma$ is small, the perturbed distribution becomes tightly concentrated around its mean, resulting in a steep gradient landscape for $-\log p_\sigma(\tilde{x})$. As $\sigma$ approaches zero, this steepness appears as sharp variations and introduces multiscale challenges. In Theorem 3.1, we provide rigorous

and refined results on this singularity compared to previous works. Eq. (10) shows that the score function explodes entirely due to its normal direction, represented by the first term on the right hand side. After subtracting this term, the remaining component is of order $O(1)$. Furthermore, for points on the manifold, Equations (11) and (12) establish a novel connection between the Riemannian score function $\nabla_x^{\mathcal{M}} \log p_0(x)$ and the perturbed score function in the ambient space. To the best of our knowledge, this connection has not been considered in previous works. The proof of Theorem 3.1 is provided in Appendix A, and the assumption of uniform boundedness of the derivative is discussed in Appendix F.3.

**Theorem 3.1.** *Let $P(x) \in \mathbb{R}^{n \times n}$ denote the projection matrix at $x \in \mathcal{M}$. Assume that $\int_{\mathcal{M}} \|x\| p_0(x) \mathrm{d}\sigma_{\mathcal{M}}(x) < +\infty$ and $M_1 = \sup_{x \in \mathcal{M}} \max_{1 \le i,j,j' \le n} \left| \frac{\partial P_{ij}}{\partial x_{j'}}(x) \right| < +\infty$. The following two asymptotic expansions for $p_\sigma(\tilde{x})$ defined in (9) hold:*

1. *For $\tilde{x} \notin \mathcal{M}$, assume that $x^* \in \mathcal{M}$ is the unique minimizer of $\min_{x \in \mathcal{M}} \|x - \tilde{x}\|$, and that $\|x^* - \tilde{x}\|_\infty < \frac{1}{n^2 M_1}$. Under these conditions, as $\sigma \to 0$, we have*

$$\nabla_{\tilde{x}} \log p_\sigma(\tilde{x}) = \frac{x^* - \tilde{x}}{\sigma^2} + O(1). \tag{10}$$

2. *For $\tilde{x} \in \mathcal{M}$, as $\sigma \to 0$, we have*

$$\nabla_{\tilde{x}} \log p_\sigma(\tilde{x}) = \nabla_{\tilde{x}}^{\mathcal{M}} \log p_0(\tilde{x}) - \frac{1}{2} \sum_{j,j'=1}^{n} \frac{\partial P_{\cdot j}}{\partial x_{j'}}(\tilde{x}) P_{jj'}(\tilde{x}) + O(\sigma), \tag{11}$$

$$\nabla_{\tilde{x}}^{\mathcal{M}} \log p_\sigma(\tilde{x}) = \nabla_{\tilde{x}}^{\mathcal{M}} \log p_0(\tilde{x}) + O(\sigma), \tag{12}$$

*where $\frac{\partial P_{\cdot j}}{\partial x_{j'}}$ denotes the vector whose $i$th component is $\frac{\partial P_{ij}}{\partial x_{j'}}$.*

Next, based on Theorem 3.1, we analyze the scale discrepancy of the score function between its tangential and normal components. To achieve this, notice that the projection matrix $P$ in (8) can be extended to the entire space $\mathbb{R}^n$, as long as $\xi$ is well-defined and smooth on $\mathbb{R}^n$. With this extended projection matrix, we further define the tangential and normal components of the score function for any $\tilde{x} \in \mathbb{R}^n$ as $P(\tilde{x}) \nabla_{\tilde{x}} \log p_\sigma(\tilde{x})$ and $P^\perp(\tilde{x}) \nabla_{\tilde{x}} \log p_\sigma(\tilde{x})$, respectively, where we denote $P^\perp(\tilde{x}) = I - P(\tilde{x})$. Using this extended definition, we then decompose the quadratic loss $\ell_{\text{quad}}$ into two parts: $\ell_{\text{quad}}^{\|}$ and $\ell_{\text{quad}}^{\perp}$, corresponding to the tangential and normal components, respectively.

$$\begin{aligned}
\ell_{\text{quad}} &= \mathbb{E}_{\tilde{x}} \|s_\theta(\tilde{x}, t) - \nabla_{\tilde{x}} \log p_\sigma(\tilde{x})\|^2 \\
&= \mathbb{E}_{\tilde{x}} \|P(\tilde{x}) s_\theta(\tilde{x}, t) - P(\tilde{x}) \nabla_{\tilde{x}} \log p_\sigma(\tilde{x})\|^2 + \mathbb{E}_{\tilde{x}} \|P^\perp(\tilde{x}) s_\theta(\tilde{x}, t) - P^\perp(\tilde{x}) \nabla_{\tilde{x}} \log p_\sigma(\tilde{x})\|^2 \\
&=: \ell_{\text{quad}}^{\|} + \ell_{\text{quad}}^{\perp}.
\end{aligned} \tag{13}$$

Using Eq. (10) and the facts that $P(x^*)(x^* - \tilde{x}) = 0$, $P(\tilde{x}) = P(x^*) + O(\tilde{x} - x^*)$ and $\mathbb{E}_{\tilde{x}|x}(x^* - \tilde{x}) = O(\sigma)$, the two terms being approximated in (13) have scales of $O(1)$ and $O(1/\sigma)$, respectively, by the following two estimates:

$$\mathbb{E}_{\tilde{x}|x}[P(\tilde{x}) \nabla_{\tilde{x}} \log p_\sigma(\tilde{x})] = \mathbb{E}_{\tilde{x}|x} \left[ (P(x^*) + O(\tilde{x} - x^*)) \left( \frac{x^* - \tilde{x}}{\sigma^2} + O(1) \right) \right] = O(1), \tag{14}$$

$$\mathbb{E}_{\tilde{x}|x}[P^\perp(\tilde{x}) \nabla_{\tilde{x}} \log p_\sigma(\tilde{x})] = \mathbb{E}_{\tilde{x}|x} \left[ (P^\perp(x^*) + O(\tilde{x} - x^*)) \left( \frac{x^* - \tilde{x}}{\sigma^2} + O(1) \right) \right] = O\left(\frac{1}{\sigma}\right), \tag{15}$$

where $\mathbb{E}_{\tilde{x}|x}$ denotes the expectation with respect to $p_\sigma(\tilde{x}|x)$.

This multiscale singularity of the loss formulation poses significant challenges during training. In fact, the above analysis also applies to the loss (5), since it differs from (4) only by a constant that is independent of $\theta$. As a result, training with (5) inherently prioritizes fitting larger-scale features of the score, which are aligned with the normal component in our settings. This is because quadratic loss minimizes the average of squared errors, making it disproportionately sensitive to errors that occur in larger-scale directions. In the context of this work, the normal component pulls samples

back onto the manifold, whereas the tangential component, which is more critical, captures the data distribution on the manifold. As a result, the model fails to adequately capture finer details of the data distribution, ultimately reducing the accuracy of the generated distribution. In other words, the trained model primarily captures the manifold itself rather than the distribution on it, leading to a phenomenon known as *manifold overfitting* [24].

To address these limitations, specific methods are required during the model training phase to ensure training stability and accurately capture the intrinsic data distribution on the manifold. Based on this perspective, we propose the following two methods. The first method reduces the scale discrepancy between the tangential and normal components by modifying the structure of the noise. The second bypasses the learning of the normal component of the score function and employs a dedicated projection operator to project samples back onto the manifold.

## 4  Methods

In this section, we propose two methods to address the singularity of the score function. The first method, referred to as Niso-DM, introduces additional noise along the normal direction to mitigate its dominance. The second method, called Tango-DM, focuses only on the training of the tangential component of the score function when the noise scale $\sigma_t$ is small.

### 4.1  Niso-DM: Perturb data with non-isotropic noise

We propose a strategy to mitigate scale discrepancies by replacing the isotropic noise in the forward process of diffusion models with non-isotropic noise. The perturbed data is generated as $\tilde{x}_t = x + \sigma_t \epsilon_1 + \sigma_t^{\alpha_t} N(x)\epsilon_2$, where $x \sim p_0(x)$, $\epsilon_1 \sim \mathcal{N}(0, I_n)$, $\epsilon_2 \sim \mathcal{N}(0, I_{n-d})$, $\alpha_t \in (0,1)$, and $N(x) \in \mathbb{R}^{n \times (n-d)}$ is defined in Section 2.2. The term $\sigma_t^{\alpha_t} N(x)\epsilon_2$ represents an additional noise along the normal directions. The conditional probability density of the perturbed data is given by $p_{\sigma_t}(\tilde{x}|x) = \mathcal{N}(x, \Sigma_{\sigma_t}(x))$, where $\Sigma_{\sigma_t}(x) = \sigma_t^2 I + \sigma_t^{2\alpha_t} N(x)N(x)^T$. The following theorem establishes the relationship between $\nabla_{\tilde{x}} \log p_{\sigma_t}(\tilde{x})$ and $\nabla_{\tilde{x}} \log p_0(\tilde{x})$, as $\sigma_t \to 0$.

**Theorem 4.1.** *Let $p_\sigma(\tilde{x})$ denote the distribution under non-isotropic perturbation, defined as:*

$$p_\sigma(\tilde{x}) = (2\pi)^{-\frac{n}{2}} \int_{\mathcal{M}} p_0(x)(\det \Sigma_\sigma(x))^{-\frac{1}{2}} e^{-\frac{1}{2}(\tilde{x}-x)^T \Sigma_\sigma(x)^{-1}(\tilde{x}-x)} d\sigma_{\mathcal{M}}(x), \qquad (16)$$

*where $\Sigma_\sigma(x) = \sigma^2 I + \sigma^{2\alpha} N(x)N(x)^T$ and $\alpha \in (0,1)$. By making the same assumptions as in Theorem 3.1, and further assuming that $M_2 = \sup_{x \in \mathcal{M}} \max_{1 \le k,l,j,j' \le n} \left| \frac{\partial^2 P_{kl}}{\partial x_j \partial x_{j'}}(x) \right| < +\infty$, we have the following results for $p_\sigma(\tilde{x})$:*

1. *For $\tilde{x} \notin \mathcal{M}$, assume that $x^* \in \mathcal{M}$ is the unique minimizer of $\min_{x \in \mathcal{M}} \|x - \tilde{x}\|$, and that $\|x^* - \tilde{x}\|_\infty < \min\{1, \frac{2}{n^2(4M_1+M_2)}\}$. Under these conditions, as $\sigma \to 0$, we have*

$$\nabla_{\tilde{x}} \log p_\sigma(\tilde{x}) = \frac{x^* - \tilde{x}}{\sigma^{2\alpha}} \cdot \frac{1}{1 + \sigma^{2-2\alpha}} + O(\sigma^{(1-2\alpha)\wedge 0}), \qquad (17)$$

   *where the symbol $\wedge$ is defined as $a \wedge b = \min\{a, b\}$.*

2. *For $\tilde{x} \in \mathcal{M}$, as $\sigma \to 0$, we have*

$$\nabla_{\tilde{x}} \log p_\sigma(\tilde{x}) = \nabla_{\tilde{x}}^{\mathcal{M}} \log p_0(\tilde{x}) + O(\sigma^{(2-2\alpha)\wedge 1}). \qquad (18)$$

Eq. (17) shows that, by introducing additional noise along the normal direction, the scale of the normal component is reduced from $O(1/\sigma^2)$ to $O(1/\sigma^{2\alpha})$. Eq. (18) states that this approach does not affect the characterization of the distribution on the manifold (i.e., the tangential component). As the noise level parameter $\sigma_t$ approaches zero, the score function of the perturbed density $p_{\sigma_t}$ on the manifold converges to the Riemannian score function of $p_0$. Therefore, by employing non-isotropic noise, we can still generate samples that adhere to the original distribution of the dataset.

Noting that $\nabla_{x_t} \log p_{\sigma_t}(x_t|x) = -\Sigma_{\sigma_t}(x)^{-1}(x_t - x)$, the denoising score matching loss (5) can be written into the following form:

$$\ell_{\text{Niso}}(t, \theta) = \mathbb{E}_{x, x_t} \|s_\theta(x_t, t) + \Sigma_{\sigma_t}(x)^{-1}(x_t - x)\|^2. \qquad (19)$$

Under the settings in Section 2.2, $\Sigma_{\sigma_t}(x)^{-1}$ has a closed-form expression (See (49)). Besides, we set $\alpha_t = \log c_{\text{niso}} / \log \sigma_t$ (i.e., $c_{\text{niso}} = \sigma_t^{\alpha_t}$) in practice, where $c_{\text{niso}}$ is a fixed constant.

## 4.2 Tango-DM: Learn only the tangential component of the score function

Recall that the projection matrix $P(x)$ in (8) is well-defined in the ambient space $\mathbb{R}^n$. Accordingly, we define $s_\theta^\parallel(x,t) := P(x)s_\theta(x,t)$ for $x \in \mathbb{R}^n$, which represents the tangential component of the parametrized vector field.

As discussed in Section 3 (see (13)), the loss function (4) and (5) can be decomposed into two parts, $\ell_{\text{quad}}^\parallel$ and $\ell_{\text{quad}}^\perp$, and the singularity issue is associated with the part $\ell_{\text{quad}}^\perp$. To address this, we propose training only the tangential component of the score function using the loss $\ell_{\text{quad}}^\parallel$ when the noise scale $\sigma_t$ is sufficiently small, thereby avoiding the singularity associated with $\ell_{\text{quad}}^\perp$. To compute $\ell_{\text{quad}}^\parallel$, we introduce the following Tango loss

$$\ell_{\text{tango}}(t,\theta) := \mathbb{E}_{x,x_t}\|s_\theta^\parallel(x_t,t) - P(x_t)\nabla_{x_t}\log p_{\sigma_t}(x_t|x)\|^2. \tag{20}$$

The optimal score network $s_{\theta*}^\parallel(x,t)$ that minimizes (20) satisfies $s_{\theta*}^\parallel(x,t) = P(x)\nabla_x \log p_{\sigma_t}(x)$ (See Appendix C.1 for the proof). This result ensures the validity of the loss function in (20).

When the noise scale is smaller than a pre-selected $c_{\text{tango}}$ (i.e. $\sigma_t < c_{\text{tango}}$), the tangential component of the score function $s_\theta^\parallel(x,t)$ is trained via the Tango loss $\ell_{\text{tango}}$. On the other hand, when $\sigma_t \geq c_{\text{tango}}$, the singularity issue is less severe, allowing us to use the original denoising score matching loss (5) to train the entire score function $s_\theta(x,t)$, which includes information along the normal direction. This enables the score function to push points toward the vicinity of the manifold. The overall loss calculation is summarized in Algorithm 1 in the Appendix.

## 5 Generation

We propose two sampling methods to generate samples using the learned score function $s_\theta$. The first method, named *Reverse SDE*, is based on the standard generation process in $\mathbb{R}^n$, with an additional projection step to ensure that the samples lie on manifolds. The second method, named *Annealing SDE*, adopts a two-stage approach, where the first stage simulates the standard reverse SDE, and the second stage performs annealed Langevin dynamics constrained to manifolds. The details of these methods are provided in the sections below and are summarized in Algorithms 2 and 3 in the Appendix.

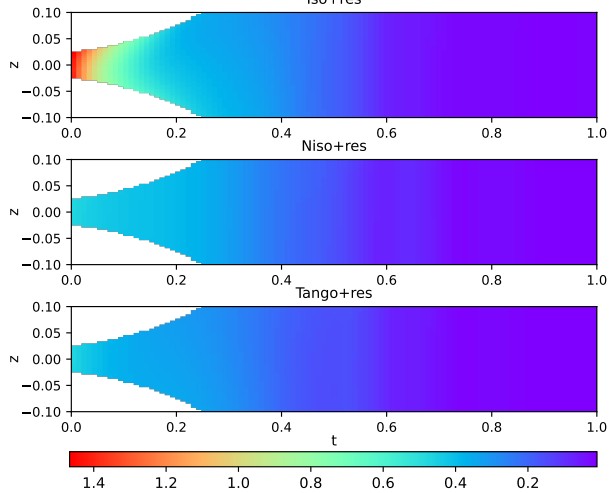

Figure 1: The average error of the tangential component of the learned score function in the $x-y$ plane, along the $z$-axis and $t$-axis, with $\sigma_{\min} = 0.01$. From top to bottom, the plots correspond to the vanilla algorithm (Iso-DM), our proposed Niso-DM and Tango-DM, all using the rescaling technique.

### 5.1 Reverse SDE

The first method follows the standard generation process, where samples are generated by simulating the reverse SDE (21) with $t$ decreasing from $T$ to 0, which is derived from (2) by setting the drift term $f$ to 0.

$$\mathrm{d}X_t = -g(t)^2 s_\theta(X_t,t)\mathrm{d}t + g(t)\mathrm{d}\overline{W}_t \tag{21}$$

However, the ending samples lie in the vicinity of the manifold rather than exactly on it. To address this, we add a final projection step to ensure that the generated points lie on the manifold.

In general, this method is not applicable to Tango-DM, which utilizes the Tango loss in (20). Since the normal component of the score function is not learned when $t$ is small, the score function $s_\theta(x,t)$

does not capture information in the normal direction. For Niso-DM, the final distribution of the reverse SDE is the same as the distribution of $x + c_{\mathrm{niso}} N(x)\epsilon_2$, where $x \sim p_0$ and $c_{\mathrm{niso}} N(x)\epsilon_2$ represents noise along the normal direction. By employing a projection onto the manifold in the subsequent step, the obtained samples can accurately approximate the target distribution.

## 5.2 Annealing SDE on manifolds

The second method adopts a two-stage sampling procedure with a predefined threshold $\tilde{\sigma}$. In the first stage ($\sigma_t \geq \tilde{\sigma}$), we simulate the reverse SDE (21), which provides a coarse approximation of the target distribution. In the second stage ($\sigma_t < \tilde{\sigma}$), we refine the samples by simulating annealed Langevin dynamics constrained to the manifold. Specifically, for a fixed $t$, we perform $n_0$ steps of the following Langevin dynamics:

$$\mathrm{d}X_s = P(X_s)s_\theta(X_s, t)\mathrm{d}s + \sqrt{2}\mathrm{d}W_s^{\mathcal{M}}. \tag{22}$$

After that, we decrease $t$ according to $t \leftarrow t - \Delta t$ and restart the simulation of SDE (22) using the final samples from the previous iteration as the initial states. This process is repeated until the desired level of refinement is achieved. i.e., $t = 0$. This stage can be viewed as the manifold version of the Corrector algorithms in [38].

To analysis the proposed sampling method, we assume that the neural network $s_\theta$ has learned the optimal solution, i.e. $P(x)s_\theta(x, t) = \nabla_x^{\mathcal{M}} \log p_{\sigma_t}(x)$ for $x \in \mathcal{M}$. Under this assumption, the invariant distribution of the SDE (22) is $p_{\sigma_t}(x)\mathrm{d}\sigma_{\mathcal{M}}(x)$. According to (12) in Theorem 3.1 and (18) in Theorem 4.1, regardless of whether non-isotropic noise is used, $\nabla_x^{\mathcal{M}} \log p_{\sigma_t}(x)$ converges to $\nabla_x^{\mathcal{M}} \log p_0(x)$ as $t \to 0$. This demonstrates that, by employing annealed Langevin dynamics constrained to the manifold, we can effectively sample from the target distribution $p_0(x)\mathrm{d}\sigma_{\mathcal{M}}(x)$ on the manifold.

# 6 Experiments

In this section, we evaluate our methods by studying several representative problems on typical manifolds. Specifically, we consider four datasets on four different types of submanifolds: (1) a hyperplane in 3D space, (2) mesh manifolds in 3D space, (3) the special orthogonal group of order 10, and (4) the level set of dihedral angles in the molecular system alanine dipeptide. Our code is available at: https://github.com/ZichenLiu1999/NisoTangoDM.

We perform experiments with the vanilla diffusion models, as well as our proposed Niso-DM and Tango-DM, denoted as *Iso*, *Niso*, and *Tango*, respectively. Moreover, we consider using neural networks to approximate normalized score functions, defined as $s_\theta(x, t) = \hat{s}_\theta(x, t)/w_t$, where $\hat{s}_\theta(x, t)$ is a neural network and $w_t$ is the scaling factor of the optimal score function. This *rescaling technique* improves numerical stability by allowing neural networks to approximate an $O(1)$ term instead of one that grows explosively. Notably, this technique is equivalent to the $\epsilon$-parameterization introduced in [33, 47]. We denote this method with the suffix +*res*. After training, new samples are generated via two methods: Reverse SDE and Annealing SDE on manifolds. Although we use "vanilla" to refer to the *Iso* and *Iso+res* methods, these two methods also involve manifold information as we utilize the projection operator during the generation process.

To evaluate the quality of the generated samples, we use several metrics to compare the generated distribution with the target distribution. The choice of metrics balances the ability to capture on-manifold distribution differences with reasonable computational cost. We conduct our experiments using the same hyperparameters, except for those specific to each algorithm. *Overall, we recommend the Niso+res training algorithm combined with the Reverse SDE generation method, as it performs best in most cases.* In experiments on mesh data and the special orthogonal group, we also conduct Riemannian sliced score matching (RSSM, for simplicity) in [8, 17] as a baseline. Further experimental details are provided in Appendix D. We also conduct ablation studies in Appendix E to analyze the bias-variance tradeoff associated with the choice of hyperparameters.

## 6.1 The hyperplane in 3D space

We first consider a toy model: the hyperplane embedded in a three-dimensional Euclidean space, specifically the set $\mathcal{M} = \{(x, y, z) \in \mathbb{R}^3 | z = 0\}$. The target distribution is a mixture of Gaussian

distributions with nine modes located on the plane. In this example, the score function has a closed-form solution that can be explicitly derived.

In Figure 1, we present the error of the tangential component of the learned score function. The results demonstrate that our proposed Niso-DM and Tango-DM significantly reduce the error of the tangential component, as $t$ approaches 0. Table 5 in the Appendix reports the maximum mean discrepancy (MMD), where both Niso-DM and Tango-DM show significant performance improvements.

## 6.2 Mesh data

We consider the Stanford Bunny [40] and Spot the Cow [7], which contain meshes derived from 3D scanning ceramic figurines of a rabbit and a cow, respectively. To create the target densities, we follow the approach in [6, 19, 32], which utilizes the eigenpairs of the Laplace-Beltrami operator on meshes. The target distribution is chosen as an equal-proportion mixture of density functions corresponding to the 0-th, 500-th, and 1000-th eigenpairs. The manifold $\mathcal{M}$ is a polyhedron composed of triangular faces, differing from the definition provided in Section 2.2. Although the manifold is not smooth along the shared edges between adjacent faces, which form a zero-measure set, numerical performance ensures that this is an acceptable setup.

Table 1 reports the Jensen–Shannon (JS) divergence of the histograms on mesh faces between the generated samples and the original datasets. With the Rescaling technique, both Niso-DM and Tango-DM demonstrate significant improvements; without it, Tango-DM still achieves improvements under the Annealing algorithm. Furthermore, we suspect that the suboptimal performance of RSSM arises from the challenges in achieving a uniform distribution on manifolds with complex geometric structures, such as the narrow neck in "Spot the Cow."

Table 1: Results for the mesh data: JS divergence of the samples generated by the Reverse SDE and Annealing SDE algorithms under different training methods. The symbol "-" indicates that the Reverse SDE sampling method is not applicable to Tango-DM. The results are reported as the mean and standard deviation over five independent runs. The suffix +*res* indicates the rescaling technique.

| | Stanford Bunny | | Spot the Cow | |
|---|---|---|---|---|
| | Reversal | Annealing | Reversal | Annealing |
| Iso | 3.58e-1±1.36e-3 | 4.95e-1±1.69e-1 | 3.41e-1±3.20e-3 | 3.65e-1±2.97e-3 |
| Niso | 3.58e-1±1.17e-3 | 4.12e-1±2.63e-3 | 3.38e-1±1.27e-3 | 3.68e-1±3.90e-3 |
| Tango | - | 4.08e-1±5.71e-4 | - | 3.51e-1±3.65e-3 |
| Iso+res | 3.52e-1±2.17e-3 | 3.94e-1±1.63e-3 | 3.30e-1±2.08e-3 | 3.48e-1±1.82e-3 |
| Niso+res | **3.48e-1**±8.27e-4 | 3.86e-1±1.85e-3 | **3.28e-1**±6.19e-4 | 3.40e-1±2.34e-3 |
| Tango+res | - | **3.84e-1**±1.57e-3 | - | **3.39e-1**±1.74e-3 |
| RSSM | 7.00e-1±3.85e-3 | | 7.22e-1±3.32e-3 | |

## 6.3 High-dimensional special orthogonal group

We evaluate the performance of our method in a high-dimensional setting on the special orthogonal group $SO(10)$, a 45-dimensional submanifold embedded in $\mathbb{R}^{100}$. We construct a synthetic dataset drawn from a multimodal distribution on $SO(10)$, consisting of 5 modes.

Our results demonstrate the effectiveness of the proposed methods. In Table 2, we compare the sliced 1-Wasserstein [2] distances between the generated and the target distributions. These results highlight the accuracy and reliability of our approach in high dimensional manifold settings. Moreover, our approach outperforms RSSM, while the vanilla algorithm does not.

## 6.4 Alanine dipeptide

We apply our method to alanine dipeptide, a model system frequently examined in computational chemistry. The system's configuration is characterized by two dihedral angles, $\phi$ and $\psi$ (refer to Figure 4). The manifold consists of the configurations of the system's 10 non-hydrogen atoms (in $\mathbb{R}^{30}$) with the angle $\phi$ fixed at $-70°$, which is a level set of the dihedral angle $\phi$.

Table 3 exhibits the results of the 2-Wasserstein distance between the test datasets and the generated samples, showing that our methods outperform the vanilla models under both the Reverse SDE and the Annealing SDE sampling algorithms. However, in this case, training the normalized score functions (+*res*) does not lead to better results. One possible explanation is that it simultaneously rescales both the tangential and normal components, resulting in the loss of critical tangential information.

Table 2: Results for $SO(10)$: Sliced 1-Wasserstein distance under different training methods. For more detailed explanations, refer to the caption of Table 1.

|  | Reversal | Annealing |
|---|---|---|
| Iso | 1.76e-2$\pm$1.15e-2 | 1.86e-2$\pm$9.65e-3 |
| Niso | 5.58e-3$\pm$1.84e-3 | 1.16e-2$\pm$2.51e-3 |
| Tango | - | 1.89e-2$\pm$2.05e-2 |
| Iso+res | 9.49e-3$\pm$1.91e-3 | 1.17e-2$\pm$7.29e-4 |
| Niso+res | **4.60e-3**$\pm$8.24e-4 | **6.00e-3**$\pm$7.53e-4 |
| Tango+res | - | 6.42e-3$\pm$1.97e-3 |
| RSSM | 7.14e-3$\pm$9.09e-4 | |

Table 3: Results for dipeptide: 2-Wasserstein distance under different training methods. For more detailed explanations, refer to the caption of Table 1.

|  | Reversal | Annealing |
|---|---|---|
| Iso | 8.60e-2$\pm$5.29e-3 | 8.83e-2$\pm$9.32e-3 |
| Niso | **8.31e-2**$\pm$5.29e-3 | **8.24e-2**$\pm$5.96e-3 |
| Tango | - | 8.60e-2$\pm$5.71e-3 |
| Iso+res | 8.59e-2$\pm$6.90e-3 | 1.21e-1$\pm$6.66e-3 |
| Niso+res | 8.34e-2$\pm$5.56e-3 | 9.47e-2$\pm$5.61e-3 |
| Tango+res | - | 9.62e-2$\pm$9.14e-3 |

# 7 Related work

Research on manifold-related diffusion models can be broadly categorized into two directions: (1) diffusion models for distributions under the manifold hypothesis, where the underlying manifold is unknown, and (2) diffusion models for distributions on a known manifold, which is the focus of this work. For the former, we discuss the singularity and the scale discrepancy of score functions mentioned in the related work; for the latter, we explore several studies based on known manifolds.

**Singularity of score functions.** Several studies on diffusion models under the manifold hypothesis [3, 27, 29] have highlighted the divergence of the score function as $t \to 0$. Beyond empirical observations, recent studies have provided mathematical analysis of diffusion models based on the VPSDE. For instance, Chen et al. [4] present a mathematical analysis under the assumption that each data point lies on a hyperplane. In [29], it is shown that the norm of the score function satisfies $\mathbb{E}\|\nabla_x \log p_t(x))\| \gtrsim 1/\sqrt{t}$. Furthermore, Lu et al. [27] rigorously prove that this singularity follows a $1/t$ scaling in a strong pointwise sense under the VPSDE, which is similar to our results in (10). Further discussion can be found in [25].

To mitigate this singularity issue, a number of works (e.g. [38]) assume that the initial time is at $t = \epsilon > 0$, instead of $t = 0$. The study [9] introduces a diffusion model on the product space of position and velocity, employing hybrid score matching to circumvent the singularity. Furthermore, in [37], the score function is parameterized by $s_\theta(x, t) = \hat{s}_\theta(x)/\sigma_t$, where $\hat{s}_\theta(x)$ is a neural network.

**Scale discrepancy of the score function.** The work [39] uses the singular value decomposition of the score matrix to identify the intrinsic dimension of the manifold, which essentially utilizes the discrepancy of the score function. References [20, 42, 44, 46] investigate the fundamental principles of diffusion models under the manifold hypothesis, by studying the eigen-decomposition of the Jacobian of the score functions. This analysis implicitly relates to the score discrepancy discussed in our work. In particular, the decomposition of the denoiser mapping (Eq. (12)) in [20], which involves a geometry-adaptive harmonic basis, also implies this discrepancy, aligning with (10) in our paper. Papers [1] and [10] study the Memorization and Generalization in Generative Diffusion under the manifold hypothesis through Hidden Manifold Models and Generalized Linear Models. Besides, [16] suggests that a conservative diffusion is guaranteed to yield the correct conclusions when analyzing local features of the data manifold.

While previous works have implicitly touched upon the scale differences in manifold settings, our work explicitly formalizes this phenomenon with rigorous mathematical formulations and introduces novel solutions to address it. Furthermore, we believe our methods can enhance the accuracy of

diffusion models during the manifold consolidation phase (as described in [42]) or the collapse transition times (as mentioned in [1, 10]), by refining the computation of the tangential component of the score function.

**Additional noise along the normal direction.**  Paper [15] proposes adding noise along the normal direction to "inflate" manifolds under the setting of normalizing flows, which is similar to our Niso method. Theorem 4 and Proposition 7 in [15] analyze the relationship between the perturbed distribution and the original distribution on the manifold, which aligns with the conclusions in our equations (12) and (18).

On the other hand, our method differs from [15] in several key aspects. In [15], the added noise has a constant magnitude, requiring additional assumptions (e.g., Q-normally reachable) to prevent interference between noise at different points. In contrast, our analysis (Theorem 3.1) permits the noise magnitude to diminish toward zero, relying on weaker assumptions. Additionally, [15] applies noise inflation prior to training the normalizing flow, whereas our method integrates noise addition directly into the diffusion process, accompanied by the noise dynamics of diffusion models. Finally, while the analysis in [15] focuses on the perturbed density function, our work studies the asymptotic behavior of the score function.

**Diffusion models on manifolds**  Riemannian Score-based Generative Models [8] extend SDE-based diffusion models to manifold settings by estimating the transition kernels using either the Riemannian logarithmic map or the eigenpairs of the Laplacian–Beltrami operator on manifolds [26]. Riemannian Diffusion Models [17] employ a variational diffusion framework for Riemannian manifolds and, similar to our approach, consider submanifolds embedded in Euclidean space. Additionally, Trivialized Diffusion Models [49] adapt diffusion models from Euclidean spaces to Lie groups.

## 8   Limitations and future work

Compared to approaches under the manifold hypothesis, our method still requires knowledge of the manifold's definition. In fields like image and language processing, the manifold structure is often assumed but not explicitly characterized. Future work will focus on extending our approach to scenarios where the manifold is undefined or implicitly represented. Furthermore, Tango-DM is not suitable for the Reverse SDE Sampling algorithm and tends to be slower due to the annealing sampler. Designing Reverse SDE-style algorithms for Tango-DM remains an open direction for future research.

## 9   Conclusion

We address the multiscale singularity of the score function for manifold-structured data, which limits the accuracy of Euclidean diffusion models. By decomposing the score function into tangential and normal components, we identify the source of scale discrepancies and propose two methods: Niso-DM, which reduces discrepancies with non-isotropic noise, and Tango-DM, which trains only the tangential component. Both methods achieve superior performance on complex manifolds, improving the accuracy and robustness of diffusion-based generative models.

## Acknowledgments and Disclosure of Funding

Zichen Liu acknowledges support from the China Scholarship Council (Grant No. 202306010047). Wei Zhang acknowledges support from DFG's Eigene Stelle (Project No. 524086759). Tiejun Li acknowledges support from the National Key R&D Program of China (Grant No. 2021YFA1003301) and the National Science Foundation of China (Grant No. 12288101).

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

# A Proof of Theorem 3.1

**Preparation.** For simplicity of notation, we use $|\cdot|$ to denote the Euclidean norm in the following two sections. We assume that $\tilde{x} \in \mathbb{R}^n$ is fixed. Let us recall the expression

$$p_\sigma(\tilde{x}) = \int_{\mathcal{M}} p_0(x) p_\sigma(\tilde{x}|x) \mathrm{d}\sigma_{\mathcal{M}}(x), \quad \sigma > 0. \tag{23}$$

Let $x^* \in \mathcal{M}$ be the unique minimizer of the minimization problem

$$\min_{x \in \mathcal{M}} |x - \tilde{x}|. \tag{24}$$

In the following, we study the limit of the integral in (23) over $\mathcal{M}$ as $\sigma \to 0$ by a change of variables using the exponential map on $\mathcal{M}$ at $x^*$. To this end, let us first derive some asymptotic expressions related to the exponential map.

Denote by $T_{x^*}\mathcal{M} = \{v|\nabla\xi(x^*)^T v = 0, v \in \mathbb{R}^n\}$ the tangent space of $\mathcal{M}$ at $x^*$. Given $v \in T_{x^*}\mathcal{M}$, let $\gamma_v(t) \in \mathbb{R}^n$ be the geodesic curve on $\mathcal{M}$ starting from $\gamma_v(0) = x^*$ at $t = 0$ such that $\dot{\gamma}_v(0) = v$. We have Taylor's expansion

$$\gamma_v(t) = x^* + vt + \frac{1}{2}\ddot{\gamma}_v(0)t^2 + O(t^3), \quad \text{as } t \to 0. \tag{25}$$

Since $x^* = \gamma_v(0)$ is the minimizer of (24), we know that $|\gamma_v(t) - \tilde{x}|^2$ (as a function of $t$) attains its minimum at $t = 0$. Therefore, taking the derivative and setting $t = 0$, we obtain

$$v \cdot (x^* - \tilde{x}) = 0, \quad \forall v \in T_{x^*}\mathcal{M}. \tag{26}$$

An expression for the second-order derivative $\ddot{\gamma}_v(0)$ in (25) is not available in general. However, by differentiating the identity $\big(I - P(\gamma_v(t))\big)\dot{\gamma}_v(t) = 0$ with respect to $t$, and setting $t = 0$, we obtain the equation (i.e. the expression for the orthogonal component of $\ddot{\gamma}_v(0)$)

$$\big(I - P(x^*)\big)\ddot{\gamma}_v(0) = \sum_{j,j'} \frac{\partial P_{\cdot j}}{\partial x_{j'}}(x^*) v_j v_{j'}, \tag{27}$$

where $\frac{\partial P_{\cdot j}}{\partial x_{j'}}(x^*)$ denotes the vector in $\mathbb{R}^n$ whose $i$-th component is $\frac{\partial P_{ij}}{\partial x_{j'}}(x^*)$, for $1 \leq i \leq n$.

The exponential map $\exp_{x^*} : T_{x^*}\mathcal{M} \to \mathcal{M}$ is well-defined in the neighbourhood $\mathcal{O}$ of $v = 0 \in T_{x*}\mathcal{M}$, and it is related to the geodesic curves by $\exp_{x^*}(v) = \gamma_{\hat{v}}(|v|)$, where $\hat{v} = v/|v|$. Therefore, from (25) we have

$$\exp_{x^*}(v) = x^* + v + \frac{|v|^2}{2}\ddot{\gamma}_{\hat{v}}(0) + O(|v|^3), \quad \text{as } |v| \to 0. \tag{28}$$

Next, let us derive an expression for $|\exp_{x^*}(v) - \tilde{x}|^2$ when $|v|$ is small. Using (26) and (28), we can obtain

$$|\exp_{x^*}(v) - \tilde{x}|^2 = |x^* - \tilde{x}|^2 + |v|^2\Big(1 + \ddot{\gamma}_{\hat{v}}(0) \cdot (x^* - \tilde{x})\Big) + O(|v|^3), \quad \text{as } |v| \to 0. \tag{29}$$

Noticing that $x^* - \tilde{x}$ is orthogonal to the tangent space $T_{x^*}\mathcal{M}$ (see (26)), using (27) we can compute the second term on the right hand side of (29)

$$|v|^2\ddot{\gamma}_{\hat{v}}(0) \cdot (x^* - \tilde{x}) = |v|^2\Big(\big(I - P(x^*)\big)\ddot{\gamma}_{\hat{v}}(0)\Big) \cdot (x^* - \tilde{x}) = \sum_{i,j,j'} \frac{\partial P_{ij}}{\partial x_{j'}}(x^*) v_j v_{j'} (x_i^* - \tilde{x}_i). \tag{30}$$

Substituting the above expression into (29) we get

$$|\exp_{x^*}(v) - \tilde{x}|^2 = |x^* - \tilde{x}|^2 + v^T S v + O(|v|^3), \quad \text{as } |v| \to 0, \tag{31}$$

where $S \in \mathbb{R}^{n \times n}$ is a matrix whose entries are

$$S_{jj'} = \delta_{jj'} + \frac{1}{2}\sum_i \Big(\frac{\partial P_{ij'}}{\partial x_j} + \frac{\partial P_{ij}}{\partial x_{j'}}\Big)(x^*)(x_i^* - \tilde{x}_i), \quad \text{for } 1 \leq j, j' \leq n. \tag{32}$$

From the Gershgorin circle theorem [41], all eigenvalues of $S$ are greater than $1 - \frac{1}{2}\max_j \sum_{i,j'} |(\frac{\partial P_{ij'}}{\partial x_j} + \frac{\partial P_{ij}}{\partial x_{j'}})(x^*)(x_i^* - \tilde{x}_i)|$. Furthermore, this lower bound is positive, as $\|x^* - \tilde{x}\|_\infty < \frac{1}{n^2 M_1}$. Therefore, the matrix $S$ is positive definite. With the above expansions, we prove the two claims.

**Proof of the first claim.** We consider the general case where $\tilde{x} \in \mathbb{R}^n$ may not belong to $\mathcal{M}$. Since $p_\sigma(\tilde{x}|x)$ is the probability density of Gaussian distribution $\mathcal{N}(x, \sigma^2 I)$, we can derive

$$
\begin{aligned}
\nabla_{\tilde{x}} \log p_\sigma(\tilde{x}) =& \frac{1}{p_\sigma(\tilde{x})} \int_{\mathcal{M}} p_0(x) \frac{x - \tilde{x}}{\sigma^2} p_\sigma(\tilde{x}|x) \mathrm{d}\sigma_{\mathcal{M}}(x) \\
=& \frac{x^* - \tilde{x}}{\sigma^2} + \frac{1}{p_\sigma(\tilde{x})} \int_{\mathcal{M}} p_0(x) \frac{x - x^*}{\sigma^2} p_\sigma(\tilde{x}|x) \mathrm{d}\sigma_{\mathcal{M}}(x) \,.
\end{aligned}
\tag{33}
$$

On the one hand, since $x^*$ is the unique minimizer of (24), there exists $\delta > 0$, such that $|x - \tilde{x}|^2 \geq |x^* - \tilde{x}|^2 + \delta$ for all $x \in \mathcal{M} \setminus \mathcal{O}'$, where $\mathcal{O}' := \exp_{x^*}(\mathcal{O})$. Therefore, we have

$$
\begin{aligned}
&\left| \int_{\mathcal{M} \setminus \mathcal{O}'} p_0(x) \frac{x - x^*}{\sigma^2} p_\sigma(\tilde{x}|x) \mathrm{d}\sigma_{\mathcal{M}}(x) \right| \\
\leq& (2\pi\sigma^2)^{-\frac{n}{2}} \left[ \int_{\mathcal{M} \setminus \mathcal{O}'} p_0(x)|x - x^*| \mathrm{d}\sigma_{\mathcal{M}}(x) \right] (\sigma^{-2} e^{-\frac{\delta}{2\sigma^2}}) e^{-\frac{|x^* - \tilde{x}|^2}{2\sigma^2}} \\
=& o(e^{-\frac{\delta}{4\sigma^2}}) e^{-\frac{|x^* - \tilde{x}|^2}{2\sigma^2}} \,.
\end{aligned}
\tag{34}
$$

On the other hand, using the fact that $\exp_{x^*}$ is a diffeomorphism on $\mathcal{O}$ with $|\det D\exp_{x^*}(v)| \equiv 1$, applying the change of variables $x = \exp_{x^*}(v)$, we get

$$
\begin{aligned}
&\int_{\mathcal{O}'} p_0(x) \frac{x - x^*}{\sigma^2} p_\sigma(\tilde{x}|x) \mathrm{d}\sigma_{\mathcal{M}}(x) \\
=& (2\pi\sigma^2)^{-\frac{n}{2}} \int_{\mathcal{O}} p_0(\exp_{x^*}(v)) \frac{\exp_{x^*}(v) - x^*}{\sigma^2} e^{-\frac{|\exp_{x^*}(v) - \tilde{x}|^2}{2\sigma^2}} \mathrm{d}v \,.
\end{aligned}
\tag{35}
$$

Using the expansion (28), we can write

$$
p_0(\exp_{x^*}(v))(\exp_{x^*}(v) - x^*) = p_0(x^*)v + \frac{1}{2}|v|^2 p_0(x^*) \ddot{\gamma}_{\hat{v}}(0) + (\nabla^{\mathcal{M}} p_0(x^*) \cdot v)v + O(|v|^3) \,.
\tag{36}
$$

Hence, using the expansions (31) and (36), we can derive the integral in (35) as

$$
\begin{aligned}
&\int_{\mathcal{O}'} p_0(x) \frac{x - x^*}{\sigma^2} p_\sigma(\tilde{x}|x) \mathrm{d}\sigma_{\mathcal{M}}(x) \\
=& \int_{\mathcal{O}} \frac{1}{\sigma^2} \left( p_0(x^*)v + (\nabla^{\mathcal{M}} p_0(x^*) \cdot v)v + \frac{1}{2}|v|^2 p_0(x^*) \ddot{\gamma}_{\hat{v}}(0) + O(|v|^3) \right) e^{-\frac{v^T S v + O(|v|^3)}{2\sigma^2}} \mathrm{d}v \\
&\cdot (2\pi\sigma^2)^{-\frac{n}{2}} e^{-\frac{|x^* - \tilde{x}|^2}{2\sigma^2}} \\
=& \int_{\mathcal{O}_\sigma} \left[ p_0(x^*) \frac{v'}{\sigma} + (\nabla^{\mathcal{M}} p_0(x^*) \cdot v')v' + \frac{1}{2}|v'|^2 p_0(x^*) \ddot{\gamma}_{\hat{v}'}(0) + O(\sigma) \right] e^{-\frac{v'^T S v' + O(\sigma)}{2}} \mathrm{d}v' \\
&\cdot (2\pi)^{-\frac{n}{2}} \sigma^{d-n} e^{-\frac{|x^* - \tilde{x}|^2}{2\sigma^2}} \\
=& O(1) \sigma^{d-n} e^{-\frac{|x^* - \tilde{x}|^2}{2\sigma^2}} \,,
\end{aligned}
\tag{37}
$$

where the second equality follows from a change of variables by $v = \sigma v'$ and $\mathcal{O}_\sigma := \{\sigma^{-1} v | v \in \mathcal{O}\}$, and the last equality follows from the fact that the integral $\int_{\mathcal{O}_\sigma} \frac{v'}{\sigma} e^{-\frac{v'^T S v' + O(\sigma)}{2}} \mathrm{d}v'$ is $O(1)$. Combining (34) and (37), we arrive at

$$
\int_{\mathcal{M}} p_0(x) \frac{x - x^*}{\sigma^2} p_\sigma(\tilde{x}|x) \mathrm{d}\sigma_{\mathcal{M}}(x) = O(1) \sigma^{d-n} e^{-\frac{|x^* - \tilde{x}|^2}{2\sigma^2}} \,.
\tag{38}
$$

Using a similar derivation, for $p_\sigma$ we can obtain the expansion

$$
p_\sigma(\tilde{x}) = \int_{\mathcal{M}} p_0(x) p_\sigma(\tilde{x}|x) \mathrm{d}\sigma_{\mathcal{M}}(x) = (2\pi\sigma^2)^{\frac{d-n}{2}} e^{-\frac{|x^* - \tilde{x}|^2}{2\sigma^2}} (p_0(x^*) + O(\sigma)) \,.
\tag{39}
$$

The first claim is obtained by combining (33), (38), and (39).

**Proof of the second claim.** Now we consider the case where $\tilde{x} \in \mathcal{M}$. First of all, notice that in this case $S = I$ and (39) simplifies to

$$p_\sigma(\tilde{x}) = (2\pi\sigma^2)^{\frac{d-n}{2}} \left(p_0(\tilde{x}) + O(\sigma)\right). \tag{40}$$

We can derive

$$
\begin{aligned}
\nabla_{\tilde{x}} p_\sigma(\tilde{x}) &= -\int_{\mathcal{M}} p_0(x) \nabla_x p_\sigma(\tilde{x}|x) \mathrm{d}\sigma_{\mathcal{M}}(x) \\
&= -\int_{\mathcal{M}} p_0(x) P(x) \nabla_x p_\sigma(\tilde{x}|x) \mathrm{d}\sigma_{\mathcal{M}}(x) - \int_{\mathcal{M}} p_0(x)(I - P(x)) \nabla_x p_\sigma(\tilde{x}|x) \mathrm{d}\sigma_{\mathcal{M}}(x) \\
&= -\int_{\mathcal{M}} p_0(x) \nabla_x^{\mathcal{M}} p_\sigma(\tilde{x}|x) \mathrm{d}\sigma_{\mathcal{M}}(x) - \int_{\mathcal{M}} p_0(x)(I - P(x)) \nabla_x p_\sigma(\tilde{x}|x) \mathrm{d}\sigma_{\mathcal{M}}(x) \\
&= \int_{\mathcal{M}} \nabla_x^{\mathcal{M}} p_0(x) p_\sigma(\tilde{x}|x) \mathrm{d}\sigma_{\mathcal{M}}(x) + \int_{\mathcal{M}} p_0(x)(I - P(x)) \frac{x - \tilde{x}}{\sigma^2} p_\sigma(\tilde{x}|x) \mathrm{d}\sigma_{\mathcal{M}}(x),
\end{aligned}
\tag{41}
$$

where $\nabla_x^{\mathcal{M}}$ denotes the gradient operator on $\mathcal{M}$ at $x$ and we used the fact that $\nabla_x^{\mathcal{M}} = P(x)\nabla_x$ to obtain the third equality, and the last equality follows by using the integration by parts formula on $\mathcal{M}$. For the first term in the last line of (41), similar to (39), we can obtain

$$\int_{\mathcal{M}} \nabla_x^{\mathcal{M}} p_0(x) p_\sigma(\tilde{x}|x) \mathrm{d}\sigma_{\mathcal{M}}(x) = (2\pi\sigma^2)^{\frac{d-n}{2}} \left(\nabla_x^{\mathcal{M}} p_0(\tilde{x}) + O(\sigma)\right). \tag{42}$$

For the second term in the last line of (41), in analogy to (35), we derive

$$
\begin{aligned}
&\int_{\mathcal{O}'} p_0(x)(I - P(x)) \frac{x - \tilde{x}}{\sigma^2} p_\sigma(\tilde{x}|x) \mathrm{d}\sigma_{\mathcal{M}}(x) \\
&= (2\pi\sigma^2)^{-\frac{n}{2}} \int_{\mathcal{O}} p_0(\exp_{\tilde{x}}(v)) \left(I - P(\exp_{\tilde{x}}(v))\right) \frac{\exp_{\tilde{x}}(v) - \tilde{x}}{\sigma^2} e^{-\frac{|\exp_{\tilde{x}}(v) - \tilde{x}|^2}{\sigma^2}} \mathrm{d}v.
\end{aligned}
\tag{43}
$$

In this case, we have $\tilde{x} \in \mathcal{M}$ and $\tilde{x} = x^*$. Therefore, the expansion (31) reduces to $|\exp_{\tilde{x}}(v) - \tilde{x}|^2 = |v|^2 + O(|v|^3)$. Applying (28), we then obtain

$$
\begin{aligned}
&(I - P(\exp_{\tilde{x}}(v)))(\exp_{\tilde{x}}(v) - \tilde{x}) \\
&= \frac{1}{2}(I - P(\tilde{x})) \ddot{\gamma}_{\hat{v}}(0) |v|^2 - \sum_{j,j'} \frac{\partial P_{\cdot j}}{\partial x_{j'}}(\tilde{x}) v_j v_{j'} + O(|v|^3) \\
&= -\frac{1}{2} \sum_{j,j'} \frac{\partial P_{\cdot j}}{\partial x_{j'}}(\tilde{x}) v_j v_{j'} + O(|v|^3),
\end{aligned}
\tag{44}
$$

where we have used (26) and (27) to derive the first and the second equality, respectively. We denote $U \in \mathbb{R}^{n \times d}$ the matrix whose columns form an orthonormal basis of $T_{\tilde{x}}\mathcal{M}$. It is straightforward to verify that $U^T U = I \in \mathbb{R}^{d \times d}$ and $P(\tilde{x}) = UU^T$. We can compute (43) as

$$
\begin{aligned}
&\int_{\mathcal{O}'} p_0(x)(I - P(x)) \frac{x - \tilde{x}}{\sigma^2} p_\sigma(\tilde{x}|x) \mathrm{d}\sigma_{\mathcal{M}}(x) \\
&= (2\pi\sigma^2)^{-\frac{n}{2}} \int_{\mathcal{O}} (p_0(\tilde{x}) + O(|v|)) \frac{1}{\sigma^2} \left(-\frac{1}{2} \sum_{j,j'} \frac{\partial P_{\cdot j}}{\partial x_{j'}}(\tilde{x}) v_j v_{j'} + O(|v|^3)\right) e^{-\frac{|v|^2 + O(|v|^3)}{2\sigma^2}} \mathrm{d}v \\
&= (2\pi)^{-\frac{n}{2}} \sigma^{d-n} \int_{\mathcal{O}_\sigma} (p_0(\tilde{x}) + O(\sigma)) \left[-\frac{1}{2} \sum_{j,j'} \frac{\partial P_{\cdot j}}{\partial x_{j'}}(\tilde{x}) v'_j v'_{j'} + O(\sigma)\right] e^{-\frac{|v'|^2 + O(\sigma)}{2}} \mathrm{d}v' \\
&= (2\pi)^{-\frac{n}{2}} \sigma^{d-n} \int_{\mathcal{W}_\sigma} \left[-\frac{1}{2} p_0(\tilde{x}) \sum_{j,j'} \frac{\partial P_{\cdot j}}{\partial x_{j'}}(\tilde{x}) \left(\sum_{l=1}^d U_{jl} w_l\right) \left(\sum_{l'=1}^d U_{j'l'} w_{l'}\right) \right. \\
&\qquad\qquad\qquad\qquad \left. + O(\sigma)\right] e^{-\frac{|Uw|^2 + O(\sigma)}{2}} \mathrm{d}w \\
&= (2\pi)^{-\frac{n}{2}} \sigma^{d-n} \int_{\mathcal{W}_\sigma} \left[-\frac{1}{2} p_0(\tilde{x}) \sum_{j,j'} \frac{\partial P_{\cdot j}}{\partial x_{j'}}(\tilde{x}) (UU^T)_{jj'} + O(\sigma)\right] e^{-\frac{|w|^2 + O(\sigma)}{2}} \mathrm{d}w \\
&= (2\pi)^{-\frac{n}{2}} \sigma^{d-n} \left[-\frac{1}{2} p_0(\tilde{x}) \sum_{j,j'} \frac{\partial P_{\cdot j}}{\partial x_{j'}}(\tilde{x}) P_{jj'}(\tilde{x}) + O(\sigma)\right],
\end{aligned}
\tag{45}
$$

where the third equality follows from a further change of variables with $v' = Uw$ and $\mathcal{W}_\sigma = \{\sigma^{-1}U^T v | v \in \mathcal{O}\}$, the fourth equality follows from $|Uw|^2 = |w|^2$ and the fact that the integral converges to an integral with respect to a standard Gaussian density. Combining (41), (42), and (45), and using the fact that the corresponding integral on $\mathcal{M} \setminus \mathcal{O}'$ is $o(e^{-\frac{\delta}{4\sigma^2}})$, we derive (11) from

$$\nabla_{\tilde{x}} \log p_\sigma(\tilde{x}) = \frac{\nabla_{\tilde{x}} p_\sigma(\tilde{x})}{p_\sigma(\tilde{x})} = \nabla_{\tilde{x}}^{\mathcal{M}} \log p_0(\tilde{x}) - \frac{1}{2}\sum_{j,j'} \frac{\partial P_{\cdot j}}{\partial x_{j'}}(\tilde{x}) P_{jj'}(\tilde{x}) + O(\sigma). \tag{46}$$

Finally, define $T(\tilde{x}) = \sum_{j,j'=1}^n \frac{\partial P_{\cdot j}}{\partial x_{j'}}(\tilde{x}) P_{jj'}(\tilde{x})$, and we have

$$
\begin{aligned}
T(\tilde{x}) &= \sum_{j,j'} \frac{\partial P_{\cdot j}}{\partial x_{j'}} P_{jj'} = \sum_{i,j,j'} \frac{\partial(P_{\cdot i}P_{ij})}{\partial x_{j'}} P_{jj'} \\
&= \sum_{i,j,j'} \frac{\partial P_{\cdot i}}{\partial x_{j'}} P_{ij} P_{jj'} + P_{\cdot i} \frac{\partial P_{ij}}{\partial x_{j'}} P_{jj'} = T(\tilde{x}) + P(\tilde{x})T(\tilde{x}).
\end{aligned}
\tag{47}
$$

This implies that $P(\tilde{x})T(\tilde{x}) = 0$. Therefore, multiplying $P(\tilde{x})$ on both sides of (46) confirms (12). The second claim is obtained.

## B   Proof of Theorem 4.1

**Preparation.**   We first study the properties of $\Sigma_\sigma(x)$. Recall that for $x \in \mathcal{M}$,

$$\Sigma_\sigma(x) = \sigma^2 I + \sigma^{2\alpha} \nabla\xi(x)\left(\nabla\xi(x)^T \nabla\xi(x)\right)^{-1} \nabla\xi(x)^T = \sigma^2 I + \sigma^{2\alpha} N(x)N(x)^T. \tag{48}$$

By the Sherman–Morrison–Woodbury formula [31],

$$
\begin{aligned}
\Sigma_\sigma(x)^{-1} &= \frac{1}{\sigma^2}\left(I - \frac{1}{1+\sigma^{2-2\alpha}}N(x)N(x)^T\right) \\
&= \frac{1}{\sigma^2}P(x) + \frac{1}{\sigma^{2\alpha}}\frac{1}{1+\sigma^{2-2\alpha}}(I - P(x)).
\end{aligned}
\tag{49}
$$

Besides,

$$\det\Sigma_\sigma(x) = \sigma^{2d}(\sigma^{2\alpha})^{n-d}(1+\sigma^{2-2\alpha})^{n-d}. \tag{50}$$

**Proof of the first claim.**   We first estimate the order of the following quadratic form:

$$
\begin{aligned}
&(x-\tilde{x})^T \Sigma_\sigma(x)^{-1}(x-\tilde{x}) \\
&= \frac{1}{\sigma^2}(x-\tilde{x})^T P(x)(x-\tilde{x}) + \frac{1}{\sigma^{2\alpha}}\frac{1}{1+\sigma^{2-2\alpha}}(x-\tilde{x})^T(I-P(x))(x-\tilde{x}).
\end{aligned}
\tag{51}
$$

We apply the same change of variables $x = \exp_{x^*}(v)$ as in the proof in Section A and $x = x^* + v + O(|v|^2)$. We obtain

$$
\begin{aligned}
&(x-\tilde{x})^T(I-P(x))(x-\tilde{x}) \\
&= \left(x^* - \tilde{x} + v + O(|v|^2)\right)^T \left(I - P(x^*) - \sum_k \frac{\partial P(x^*)}{\partial x_k}v_k + O(|v|^2)\right)\left(x^* - \tilde{x} + v + O(|v|^2)\right) \\
&= |x^* - \tilde{x}|^2 + O(|v|^2),
\end{aligned}
\tag{52}
$$

and

$$(x - \tilde{x})^T P(x)(x - \tilde{x})$$

$$= \left( x^* - \tilde{x} + v + O(|v|^2) \right)^T \left( P(x^*) + \sum_k \frac{\partial P(x^*)}{\partial x_k} v_k + \frac{1}{2} \sum_{k,l} \frac{\partial^2 P(x^*)}{\partial x_k \partial x_l} v_k v_l + O(|v|^3) \right)$$

$$\cdot \left( x^* - \tilde{x} + v + O(|v|^2) \right)$$

$$= |v|^2 + 2 \sum_{k,j,j'} \frac{\partial P_{jj'}}{\partial x_k}(x^*) v_k v_{j'}(x_j^* - \tilde{x}_j) + \frac{1}{2} \sum_{k,l,j,j'} \frac{\partial^2 P(x^*)}{\partial x_k \partial x_l} v_k v_l (x_j^* - \tilde{x}_j)(x_{j'}^* - \tilde{x}_{j'}) + O(|v|^3)$$

$$:= v^T \tilde{S} v + O(|v|^3).$$

(53)

The disappearance of the first-order terms in (52) and (53) is attributed to

$$(x^* - \tilde{x})^T \left( \sum_k \frac{\partial P(x^*)}{\partial x_k} v_k \right) (x^* - \tilde{x})$$

$$= (x^* - \tilde{x})^T \left( \sum_k \frac{\partial (P)^2(x^*)}{\partial x_k} v_k \right) (x^* - \tilde{x})$$

$$= (x^* - \tilde{x})^T P(x^*)^T \left( \sum_k \frac{\partial P(x^*)}{\partial x_k} v_k \right) (x^* - \tilde{x}) + (x^* - \tilde{x})^T \left( \sum_k \frac{\partial P(x^*)}{\partial x_k} v_k \right) P(x^*)(x^* - \tilde{x})$$

$$= 0,$$

(54)

and $\tilde{S} \in \mathbb{R}^{n \times n}$ is a matrix whose entries are defined by

$$\tilde{S}_{jj'} = \delta_{jj'} + \sum_i \left( \frac{\partial P_{ij'}}{\partial x_j} + \frac{\partial P_{ij}}{\partial x_{j'}} \right)(x^*)(x_i^* - \tilde{x}_i) + \frac{1}{2} \sum_{k,l} \frac{\partial^2 P_{kl}(x^*)}{\partial x_j \partial x_{j'}}(x_k^* - \tilde{x}_l)(x_k^* - \tilde{x}_l), \quad (55)$$

for $1 \leq j, j' \leq n$. By the Gershgorin circle theorem [41] and the condition $\|x^* - \tilde{x}\|_\infty < \min\{1, \frac{2}{n^2(4M_1 + M_2)}\}$, the matrix $\tilde{S}$ is also positive definite. From (52), (53), and (55), $p_\sigma(\tilde{x}|x)$ can be written as

$$p_\sigma(\tilde{x}|x) = (2\pi)^{-\frac{n}{2}} \sigma^{-d} (\sigma^{-\alpha})^{n-d} (1 + o(1)) e^{-\frac{1}{2\sigma^2} v^T \tilde{S} v - \frac{1}{\sigma^{2\alpha}} |x^* - \tilde{x}|^2 + \frac{1}{\sigma^2} O(|v|^3) + \frac{1}{\sigma^{2\alpha}} O(|v|^2)}. \quad (56)$$

Noting that $(x - \tilde{x})^T \Sigma_\sigma(x)^{-1}(x - \tilde{x}) \geq \frac{1}{1+\sigma^{2-2\alpha}} \frac{1}{\sigma^{2\alpha}} |x - \tilde{x}|^2$, there exists $\delta > 0$ such that $(x - \tilde{x})^T \Sigma_\sigma(x)^{-1}(x - \tilde{x}) - \frac{1}{\sigma^{2\alpha}} |x^* - \tilde{x}|^2 \geq \frac{\delta}{\sigma^{2\alpha}}$ for all $x \in \mathcal{M} \setminus \mathcal{O}'$, where $\mathcal{O}' := \exp_{x^*}(\mathcal{O})$. For any $m \in \mathbb{R}$, we have

$$\sigma^m \int_{\mathcal{M}} |x - x^*| p_0(x) p_\sigma(\tilde{x}|x) d\sigma_{\mathcal{M}}(x)$$

$$\leq \sigma^m e^{-\frac{1}{\sigma^{2\alpha}} |x^* - \tilde{x}|^2 - \frac{\delta}{\sigma^{2\alpha}}} \int_{\mathcal{M}} (2\pi)^{-\frac{n}{2}} |\det \Sigma_\sigma(x)|^{-\frac{1}{2}} |x - x^*| p_0(x) d\sigma_{\mathcal{M}}(x) \qquad (57)$$

$$\leq e^{-\frac{1}{\sigma^{2\alpha}} |x^* - \tilde{x}|^2} o(e^{-\frac{\delta}{4\sigma^\alpha}}).$$

Similarly, the order of $-\Sigma_\sigma(x)^{-1}(\tilde{x} - x)$ can be estimated as

$$- \Sigma_\sigma(x)^{-1}(\tilde{x} - x)$$

$$= \frac{1}{\sigma^2} P(x)(x - \tilde{x}) + \frac{1}{\sigma^{2\alpha}} \frac{1}{1 + \sigma^{2-2\alpha}} (I - P(x))(x - \tilde{x})$$

$$= \frac{1}{\sigma^2} \left( P(x^*) + \sum_k \frac{\partial P}{\partial x_k}(x^*) v_k + O(|v|^2) \right) (x^* - \tilde{x} + v + O(|v|^2)) \qquad (58)$$

$$+ \frac{1}{\sigma^{2\alpha}} \frac{1}{1 + \sigma^{2-2\alpha}} \left( I - P(x^*) - \sum_k \frac{\partial P}{\partial x_k}(x^*) v_k + O(|v|^2) \right) (x^* - \tilde{x} + v + O(|v|^2))$$

$$= \frac{1}{\sigma^2} (v + Hv + O(|v|^2)) + \frac{1}{\sigma^{2\alpha}} \frac{1}{1 + \sigma^{2-2\alpha}} (x^* - \tilde{x} - Hv) + \frac{1}{\sigma^{2\alpha}} O(|v|^2),$$

where we denote $H \in \mathbb{R}^{n \times n}$ with $H_{ij} = \sum_l \frac{\partial P_{il}}{\partial x_j}(\tilde{x})(x_l^* - x_l)$. Therefore,

$$
\nabla_{\tilde{x}} p_\sigma(\tilde{x}) - \frac{x^* - \tilde{x}}{\sigma^{2\alpha}} \frac{1}{1 + \sigma^{2-2\alpha}} p_\sigma(\tilde{x})
$$

$$
= \int_{\mathcal{M}} p_0(x) \left( -\Sigma_\sigma(x)^{-1}(\tilde{x} - x) - \frac{x^* - \tilde{x}}{\sigma^{2\alpha}} \frac{1}{1 + \sigma^{2-2\alpha}} \right) p_\sigma(\tilde{x}|x) d\sigma_{\mathcal{M}}(x)
$$

$$
= \int_{\mathcal{O}} p_0(x) \left( \frac{1}{\sigma^2}(v + Hv + O(|v|^2)) - \frac{1}{\sigma^{2\alpha}} \frac{1}{1 + \sigma^{2-2\alpha}} Hv + \frac{1}{\sigma^{2\alpha}} O(|v|^2) \right) p_\sigma(\tilde{x}|x) d\sigma_{\mathcal{M}}(x)
$$

$$
+ e^{-\frac{1}{\sigma^{2\alpha}}|x^* - \tilde{x}|^2} o(e^{-\frac{\delta}{4\sigma^\alpha}})
$$

$$
= \int_{\mathcal{O}} \sigma^{-d} \left( p_0(x^*) + O(|v|) \right) \left( \frac{1}{\sigma^2}(v + Hv + O(|v|^2)) - \frac{1}{\sigma^{2\alpha}} \frac{1}{1 + \sigma^{2-2\alpha}} Hv + \frac{1}{\sigma^{2\alpha}} O(|v|^2) \right)
$$

$$
\cdot e^{-\frac{1}{2\sigma^2} v^T \tilde{S} v + \frac{1}{\sigma^2} O(|v|^3) + \frac{1}{\sigma^{2\alpha}} O(|v|^2)} dv \cdot (2\pi)^{-\frac{n}{2}} (\sigma^{-\alpha})^{n-d} e^{-\frac{1}{\sigma^{2\alpha}}|x^* - \tilde{x}|^2} (1 + o(1))
$$

$$
+ e^{-\frac{1}{\sigma^{2\alpha}}|x^* - \tilde{x}|^2} o(e^{-\frac{\delta}{4\sigma^\alpha}})
$$

$$
= \int_{\mathcal{O}_\sigma} \left( p_0(x^*) + O(\sigma) \right) \left( \frac{1}{\sigma}(I + H)v' - \frac{\sigma^{1-2\alpha}}{1 + \sigma^{2-2\alpha}} Hv' + O(1) \right) e^{-v'^T S_1 v' + O(\sigma) + O(\sigma^{2-2\alpha})} dv'
$$

$$
\cdot (2\pi)^{-\frac{n}{2}} (\sigma^{-\alpha})^{n-d} e^{-\frac{1}{\sigma^{2\alpha}}|x^* - \tilde{x}|^2} + e^{-\frac{1}{\sigma^{2\alpha}}|x^* - \tilde{x}|^2} o(e^{-\frac{\delta}{4\sigma^\alpha}})
$$

$$
= \sigma^{(d-n)\alpha} e^{-\frac{1}{\sigma^{2\alpha}}|x^* - \tilde{x}|^2} O(\sigma^{(1-2\alpha)\wedge 0}),
$$

(59)

where the third equality follows from a change of variables by $v = \sigma v'$ and $\mathcal{O}_\sigma := \{\sigma^{-1} v | v \in \mathcal{O}\}$, and $O(\sigma) + O(\sigma^{2-2\alpha}) = O(\sigma^{(2-2\alpha)\wedge 1})$. Besides, using a similar derivation, we can obtain the expansion for $p_\sigma$,

$$
p_\sigma(\tilde{x}) = \sigma^{(d-n)\alpha} e^{-\frac{1}{\sigma^{2\alpha}}|x^* - \tilde{x}|^2} \left( p_0(x^*) + O(\sigma) \right).
$$

(60)

The first claim is obtained by combining (59) and (60).

**Proof of the second claim.** Now we consider the case where $\tilde{x} \in \mathcal{M}$. In this case, $\tilde{x} = x^*$. By the change of variables $x = \exp_{x^*}(v)$, the quadratic form becomes $(x - \tilde{x})^T \Sigma_\sigma(x)^{-1}(x - \tilde{x}) = \frac{1}{\sigma^2}(|v|^2 + O(|v|^3)) + \frac{1}{\sigma^{2\alpha}} O(|v|^4)$. Similar to (41) and (45), we aim to show that

$$
\nabla_{\tilde{x}} p_\sigma(\tilde{x}) - \int_{\mathcal{M}} \nabla_x^{\mathcal{M}} p_0(x) p_\sigma(\tilde{x}|x) d\sigma_{\mathcal{M}}(x)
$$

$$
= (2\pi)^{-\frac{n}{2}} \sigma^{(d-n)\alpha} p_0(\tilde{x}) \left[ -\frac{1}{2} \sum_{j,j'=1}^n \nabla_{\tilde{x}} P_{jj'}(\tilde{x}) P_{jj'}(\tilde{x}) + O(\sigma^{1 \wedge (2-2\alpha)}) \right].
$$

(61)

First, by integration by parts,

$$
\nabla_{\tilde{x}} p_\sigma(\tilde{x}) - \int_{\mathcal{M}} \nabla_x^{\mathcal{M}} p_0(x) p_\sigma(\tilde{x}|x) d\sigma_{\mathcal{M}}(x)
$$

$$
= \int_{\mathcal{M}} p_0(x) \left( \nabla_{\tilde{x}} p_\sigma(\tilde{x}|x) + \nabla_x^{\mathcal{M}} p_\sigma(\tilde{x}|x) \right) d\sigma_{\mathcal{M}}(x)
$$

$$
= \int_{\mathcal{M}} p_0(x) \left[ (I - P(x)) \Sigma_\sigma(x)^{-1}(x - \tilde{x}) - \frac{1}{2} \sum_{i,j,k} (x_i - \tilde{x}_i) P_{\cdot k} \frac{\partial (\Sigma_\sigma^{ij})^{-1}}{\partial x_k}(x)(x_j - \tilde{x}_j) \right]
$$

$$
\cdot p_\sigma(\tilde{x}|x) d\sigma_{\mathcal{M}}(x),
$$

(62)

where $\Sigma_\sigma^{ij}$ refers to the $(i,j)$ entry of the matrix $\Sigma_\sigma$. From (49), we have

$$
(I - P(x)) \Sigma_\sigma(x)^{-1}(x - \tilde{x}) = \frac{1}{\sigma^{2\alpha}} \frac{1}{1 + \sigma^{2-2\alpha}} (I - P(x))(x - \tilde{x}) = \frac{1}{\sigma^{2\alpha}} O(|v|^2)
$$

(63)

and

$$\frac{1}{2}\sum_{i,j,k}(x_i-\tilde{x}_i)P_{\cdot k}\frac{\partial}{\partial x_k}(\Sigma_\sigma^{ij})^{-1}(x)(x_j-\tilde{x}_j)$$

$$=\frac{1}{2\sigma^2}\sum_{i,j,k}(x_i-\tilde{x}_i)P_{\cdot k}\frac{\partial}{\partial x_k}P_{ij}(x)(x_j-\tilde{x}_j)+\frac{1}{\sigma^{2\alpha}}O(|v|^2) \tag{64}$$

$$=\frac{1}{2\sigma^2}\sum_{i,j,k}P_{\cdot k}\frac{\partial P_{ij}}{\partial x_k}(\tilde{x})v_iv_j+\frac{1}{2\sigma^2}O(|v|^3)+\frac{1}{\sigma^{2\alpha}}O(|v|^2).$$

Next, we continue to calculate (62) using (63) and (64). There exists $\delta>0$ and $\mathcal{O}'$ such that

$$(x-\tilde{x})^T\Sigma_\sigma(x)^{-1}(x-\tilde{x})\geq\frac{1}{2\sigma^{2\alpha}}\frac{|x-\tilde{x}|^2}{1+\sigma^{2-2\alpha}}>\frac{\delta}{\sigma^{2\alpha}},\ \forall\sigma>0,\ x\in\mathcal{M}\setminus\mathcal{O}'. \tag{65}$$

Therefore, the value of the integral (62) in $x\in\mathcal{M}\setminus\mathcal{O}'$ is $o(e^{-\frac{\delta}{4\sigma^\alpha}})$. Using the same derivation as in (45), we have

$$\nabla_{\tilde{x}}p_\sigma(\tilde{x})-\int_{\mathcal{M}}\nabla_x^{\mathcal{M}}p_0(x)p_\sigma(\tilde{x}|x)\mathrm{d}\sigma_{\mathcal{M}}(x)$$

$$=\int_{\mathcal{O}'}p_0(\tilde{x}+O(|v|))\left(-\frac{1}{2\sigma^2}\sum_{i,j,k}P_{\cdot k}\frac{\partial P_{ij}}{\partial x_k}(\tilde{x})v_iv_j+\frac{1}{2\sigma^2}O(|v|^3)+\frac{1}{\sigma^{2\alpha}}O(|v|^2)\right)$$ 

$$\cdot p_\sigma(\tilde{x}|\exp_{x^*}(v))\mathrm{d}v+o(e^{-\frac{\delta}{4\sigma^\alpha}}) \tag{66}$$

$$=(2\pi)^{-\frac{n}{2}}\sigma^{(d-n)\alpha}p_0(\tilde{x})\Big[-\frac{1}{2}\sum_{k,j,j'}P_{\cdot k}\frac{\partial P_{jj'}}{\partial x_k}(\tilde{x})P_{jj'}(\tilde{x})+O(\sigma^{1\wedge(2-2\alpha)})\Big]+o(e^{-\frac{\delta}{4\sigma^\alpha}}).$$

Therefore, (61) holds. Besides,

$$\sum_{j,j'}\nabla P_{jj'}P_{jj'}=\sum_{k,j,j'}\nabla(P_{jk}P_{kj'})P_{jj'}$$

$$=\sum_{k,j,j'}\nabla P_{kj'}P_{jk}P_{jj'}+\sum_{k,j,j'}\nabla P_{jk}P_{kj'}P_{jj'} \tag{67}$$

$$=2\sum_{j,j'}\nabla P_{jj'}P_{jj'}$$

implies $\sum_{j,j'}\nabla P_{jj'}(\tilde{x})P_{jj'}(\tilde{x})=0$. Therefore, (61) becomes

$$\nabla_{\tilde{x}}p_\sigma(\tilde{x})-\int_{\mathcal{M}}\nabla_x^{\mathcal{M}}p_0(x)p_\sigma(\tilde{x}|x)\mathrm{d}\sigma_{\mathcal{M}}(x)=(2\pi)^{-\frac{n}{2}}\sigma^{(d-n)\alpha}p_0(\tilde{x})O(\sigma^{1\wedge(2-2\alpha)}). \tag{68}$$

Similarly, we have

$$\int_{\mathcal{M}}\nabla_x^{\mathcal{M}}p_0(x)p_\sigma(\tilde{x}|x)\mathrm{d}\sigma_{\mathcal{M}}(x)=(2\pi)^{-\frac{n}{2}}\sigma^{(d-n)\alpha}\left(\nabla_{\tilde{x}}^{\mathcal{M}}p_0(\tilde{x})+O(\sigma^{1\wedge(2-2\alpha)})\right), \tag{69}$$

$$p_\sigma(\tilde{x})=\int_{\mathcal{M}}p_0(x)p_\sigma(\tilde{x}|x)\mathrm{d}\sigma_{\mathcal{M}}(x)=(2\pi)^{-\frac{n}{2}}\sigma^{(d-n)\alpha}\left(p_0(\tilde{x})+O(\sigma^{1\wedge(2-2\alpha)})\right). \tag{70}$$

Using (68), (69), and (70), the second claim is obtained via

$$\nabla_{\tilde{x}}\log p_\sigma(\tilde{x})=\frac{\nabla_{\tilde{x}}p_\sigma(\tilde{x})}{p_\sigma(\tilde{x})}=\nabla_{\tilde{x}}^{\mathcal{M}}\log p_0(\tilde{x})+O(\sigma^{1\wedge(2-2\alpha)}). \tag{71}$$

## C   Algorithm details

Figure 2 illustrates the schematic diagrams of the vanilla method, along with our proposed Niso-DM and tango-DM.

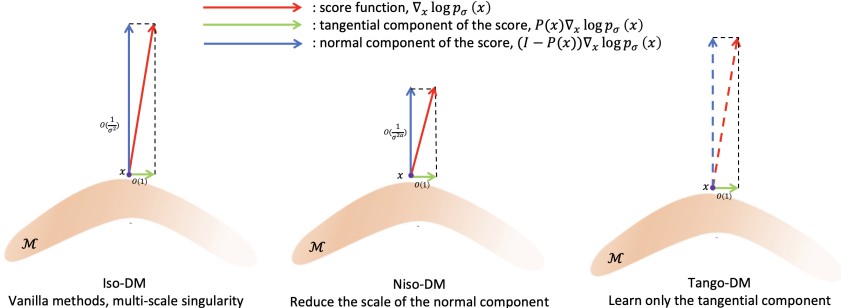

Figure 2: Schematic diagrams of the vanilla method (left), our proposed Niso-DM (middle), and tango-DM (right).

### C.1 Details of the Tango loss

We first prove the validity of the loss (20). Note that

$$
\begin{aligned}
&\mathbb{E}_{x_t}\|s_\theta^{\|}(x_t,t) - P(x_t)\nabla_{x_t}\log p_{\sigma_t}(x_t)\|^2\\
=&\mathbb{E}_{x_t}\|P(x_t)s_\theta(x_t,t)\|^2 - 2\mathbb{E}_{x_t}\langle P(x_t)s_\theta(x_t,t), P(x_t)\nabla_{x_t}\log p_{\sigma_t}(x_t)\rangle + C\\
=&\mathbb{E}_{x_t}\|P(x_t)s_\theta(x_t,t)\|^2 - 2\int_{\mathbb{R}^n}\langle P(x_t)s_\theta(x_t,t), \nabla_{x_t}p_{\sigma_t}(x_t)\rangle\mathrm{d}x_t + C\\
=&\mathbb{E}_{x_t}\|P(x_t)s_\theta(x_t,t)\|^2 - 2\int_{\mathbb{R}^n}\left\langle P(x_t)s_\theta(x_t,t), \nabla_{x_t}\int_{\mathcal{M}}p_{\sigma_t}(x_t|x)p_0(x)\mathrm{d}x\right\rangle\mathrm{d}x_t + C\\
=&\mathbb{E}_{x_t}\|P(x_t)s_\theta(x_t,t)\|^2 - 2\int_{\mathbb{R}^n}\int_{\mathcal{M}}\langle P(x_t)s_\theta(x_t,t), \nabla_{x_t}\log p_{\sigma_t}(x_t|x)\rangle\, p_{\sigma_t}(x_t|x)p_0(x)\mathrm{d}x\mathrm{d}x_t + C\\
=&\mathbb{E}_{x_t}\|P(x_t)s_\theta(x_t,t)\|^2 - 2\mathbb{E}_{x,x_t}\langle P(x_t)s_\theta(x_t,t), \nabla_{x_t}\log p_{\sigma_t}(x_t|x)\rangle + C\\
=&\mathbb{E}_{x,x_t}\|s_\theta^{\|}(x_t,t) - P(x_t)\nabla_{x_t}\log p_{\sigma_t}(x_t|x)\|^2 + C_1\,,
\end{aligned}
\tag{72}
$$

where $C$ and $C_1$ are constants independent of $\theta$. Therefore, the minimizer of the loss (20) satisfies that $s_\theta^{\|}(x,t) = P(x)\nabla_x\log p_{\sigma_t}(x)$. The overall loss calculation is summarized in Algorithm 1.

---

**Algorithm 1** The overall loss calculation with the Tango loss

---

**Require:** neural network $s_\theta$, threshold $c_{\text{tango}}$
1: Sample $t \sim \mathcal{U}(0,1)$, $x \sim p_0$
2: **if** $\sigma_t \geq c_{\text{tango}}$ **then**
3:     Calculate loss: $\lambda_t\mathbb{E}_{x,x_t}\|s_\theta(x_t,t) - \nabla_{x_t}\log p_{\sigma_t}(x_t|x)\|^2$
4: **else**
5:     Calculate loss: $\lambda_t\mathbb{E}_{x,x_t}\|P(x_t)s_\theta(x_t,t) - P(x_t)\nabla_{x_t}\log p_{\sigma_t}(x_t|x)\|^2$
6: **end if**

---

### C.2 Details of the sampling algorithm

Algorithms 2 and 3 provide detailed descriptions of the Reverse SDE and the Annealing SDE on manifolds, respectively. The threshold $\tilde\sigma$ in Algorithm 3 is set to $c_{\text{niso}}$ and $c_{\text{tango}}$ for Niso-DM and Tango-DM, respectively.

## D Experimental details

The details of each experiment are provided in the subsections below. The neural networks used are multilayer perceptrons (MLPs) with SiLU activations. Models are trained using PyTorch, utilizing the Adam optimizer with a fixed learning rate, and gradients are clipped when their 2-norm exceeds a predefined threshold. An exponential moving average of the model weights [30] is applied with a

---

**Algorithm 2** Reverse SDE Solver

---

**Require:** trained neural network $s_\theta$, total number of discrete SDE steps $N$, projection operator $\pi$
1:   $t \leftarrow T, \Delta t = T/N$
2:   $x_t \sim \mathcal{N}(0, \sigma_{\max}^2 I)$
3:   **while** $t > 0$ **do**
4:     $z \sim \mathcal{N}(0, I)$
5:     $x_t \leftarrow x_t + g(t)^2 s_\theta(x_t, t) \Delta t + g(t) \sqrt{\Delta t} z$
6:     $t \leftarrow t - \Delta t$
7:   **end while**
8:   *%Projection onto the manifold*
9:   $x_t \leftarrow \pi(x_t)$
10: **return** $x_t$

---

---

**Algorithm 3** Annealing SDE on manifolds

---

**Require:** trained neural network $s_\theta$, total number of discrete SDE steps $N$, threshold $\tilde{\sigma}$, number of steps $n_0$ and step size $\alpha_{\text{ld}}$ for the Langevin dynamics on manifolds, projection operator $\pi$
1:   $t \leftarrow T, \Delta t = T/N, x_t \sim \mathcal{N}(0, \sigma_{\max}^2 I)$
2:   *%Stage 1: Reverse SDE*
3:   **while** $\sigma_t \geq \tilde{\sigma}$ **do**
4:     $z \sim \mathcal{N}(0, I)$
5:     $x_t \leftarrow x_t + g(t)^2 s_\theta(x_t, t) \Delta t + g(t) \sqrt{\Delta t} z$
6:     $t \leftarrow t - \Delta t$
7:   **end while**
8:   *%Stage 2: Annealing SDE on manifolds*
9:   $x_t \leftarrow \pi(x_t)$
10: **while** $t > 0$ **do**
11:    **for** $i = 0$ to $n_0$ **do**
12:      $z \sim \mathcal{N}(0, I), z_1 \leftarrow P(x)z$
13:      $x_t' \leftarrow x_t + \alpha_{\text{ld}} P(x_t) s_\theta(x_t, t) + \sqrt{2\alpha_{\text{ld}}} z_1$
14:      $x_t \leftarrow \pi(x_t')$
15:    **end for**
16:    $t \leftarrow t - \Delta t$
17: **end while**
18: **return** $x_t$

---

decay rate of $0.999$. In each run, the dataset is divided into training and test sets with a ratio $8 : 2$. All experiments are conducted on either a single Tesla V100-PCIE-32GB GPU or an NVIDIA A40 GPU with 48 GB of memory. The values of all parameters used in the experiments are listed in Table 4.

When the rescaling technique is not applied, the time reweighting coefficient $\lambda_t$ is defined as $\sigma_t^2$. In contrast, when the rescaling technique is applied, $\lambda_t$ is defined as $\sigma_t w_t$, where $w_t$ represents the scaling factor of the optimal score function. Specifically, $w_t$ corresponds to $\sigma_t$ in Iso-DM, $\sqrt{\sigma_t^2 + c_{\text{niso}}^2}$ in Niso-DM, and $\max\{\sigma_t, c_{\text{tango}}\}$ in Tango-DM.

## D.1   Riemannian sliced score matching

Diffusion models constrained to a manifold often rely on complex geometric constructs, such as the heat kernel or logarithmic mapping, to compute transition probabilities. These dependencies limit the efficient computation of the denoising score matching loss. In contrast, Riemannian sliced score matching (RSSM [8, 17]) offers broader applicability to general manifolds. This is achieved by leveraging the implicit score matching loss [8], combined with Hutchinson trace estimation [18]. The corresponding training loss is expressed as:

$$\ell_{\text{RSSM}}(t, \theta) := \mathbb{E}_{x_t} \|s_\theta(x_t, t)\|^2 + 2\mathbb{E}_{z \sim \mathcal{N}(0, I), x_t} \left[ z^T P(x_t)^T \nabla_{x_t} s_\theta(x_t, t) P(x_t) z \right]. \qquad (73)$$

Table 4: Parameters in our experiments. $\sigma_{\min}$, $\sigma_{\max}$, and $T$ are the parameters of the SDE. $c_{\text{niso}}$ and $c_{\text{tango}}$ are the parameters for Niso-DM and Tango-DM, respectively. $N$, $n_0$, and $\alpha_{\text{ld}}$ are the hyperparameters of the sampling algorithm. $N_{\text{epoch}}$, $B$, lr, and clip denote the total epochs, the batch size, the learning rate, and the gradient clipping threshold during training. $N_{\text{node}}$ and $N_{\text{layer}}$ represent the number of nodes per layer and the number of hidden layers in the neural networks, respectively. $\mathcal{D}$ indicates the dataset size.

| Parameters | Hyperplane | Bunny | Spot | SO(10) | dipeptide |
|---|---|---|---|---|---|
| $\sigma_{\min}$ | 0.001 | 0.001 | 0.001 | 0.0005 | 0.0001 |
| $\sigma_{\max}$ | 3 | 3 | 3 | 3 | 5 |
| $T$ | 1 | 1 | 1 | 1 | 1 |
| $c_{\text{niso}}$ | 0.2 | 0.002 | 0.002 | 0.01 | 0.005 |
| $c_{\text{tango}}$ | 0.2 | 0.002 | 0.002 | 0.05 | 0.005 |
| $N$ | 500 | 200 | 200 | 500 | 500 |
| $n_0$ | 10 | 20 | 10 | 10 | 10 |
| $\alpha_{\text{ld}}$ | 0.01 | 0.05 | 0.05 | 0.05 | 0.05 |
| $N_{\text{node}}$ | 64 | 256 | 256 | 512 | 512 |
| $N_{\text{layer}}$ | 3 | 3 | 3 | 3 | 5 |
| $N_{\text{epoch}}$ | 200 | 20000 | 20000 | 5000 | 6000 |
| $B$ | 512 | 4096 | 4096 | 512 | 1024 |
| lr | 0.0005 | 0.0005 | 0.0005 | 0.001 | 0.0005 |
| clip | 10 | 10 | 10 | 1 | 10 |
| $\mathcal{D}$ | 50000 | 60000 | 60000 | 50000 | 99999 |

Table 5: Results for the Hyperplane: MMD under different training methods. In this example, the Reverse SDE sampling method is applicable to Tango-DM due to the complete decoupling of tangential and normal components. For more detailed explanations, refer to the caption of Table 1.

| | Reversal | Annealing |
|---|---|---|
| Iso | 2.81e-4$_{\pm 5.66e\text{-}5}$ | 7.97e-4$_{\pm 6.89e\text{-}5}$ |
| Niso | 1.52e-4$_{\pm 2.60e\text{-}5}$ | 5.32e-4$_{\pm 3.03e\text{-}4}$ |
| Tango | 1.65e-4$_{\pm 3.91e\text{-}5}$ | 5.82e-4$_{\pm 1.17e\text{-}4}$ |
| Iso+res | 2.79e-4$_{\pm 9.41e\text{-}5}$ | 7.81e-4$_{\pm 2.69e\text{-}4}$ |
| Niso+res | 1.45e-4$_{\pm 2.68e\text{-}5}$ | **1.88e-4**$_{\pm 4.49e\text{-}5}$ |
| Tango+res | **1.19e-4**$_{\pm 1.96e\text{-}5}$ | 2.22e-4$_{\pm 3.03e\text{-}5}$ |

### D.2 The hyperplane in 3D space

The target distribution is a mixture of Gaussian distributions with nine modes located on the plane. Specifically, the means of the nine modes are $(-1, -1)$, $(-1, 0)$, $(-1, 1)$, $(0, -1)$, $(0, 0)$, $(0, 1)$, $(1, -1)$, $(1, 0)$, and $(1, 1)$, and each Gaussian has a standard deviation of 0.3.

### D.3 Mesh data

To create the datasets, we first obtain the clamped eigenfunctions of the Laplacian operator on a mesh that has been upsampled threefold for the original mesh. The target distribution is chosen as an equal-proportion mixture of density functions corresponding to the 0-th, 500-th, and 1000-th eigenpairs. For the distribution on the cow mesh, we additionally discarded the points on the horns and tail.

For $x \notin \mathcal{M}$, the closest point $\pi(x)$ on a triangular mesh surface is determined as follows: First, $x$ is projected onto the planes of all triangular faces in the mesh using the face normals and vertex positions. Barycentric coordinates are then computed to check whether the projected points lie inside the triangles. If a point falls outside a triangle, it is further projected onto the nearest edge to ensure it remains on the triangle's boundary. Subsequently, the Euclidean distance between the original query

point $x$ and all projected points is calculated, and the closest point $\pi(x)$ is selected by identifying the minimum distance.

We assign the normal direction $n(\pi(x))$ of $\pi(x)$ as the normal direction for $x$ $(x \notin \mathcal{M})$. When $\pi(x)$ is located inside a triangle of the mesh, the normal of the triangle is chosen as the normal vector at $\pi(x)$ (i.e. $n(x) = n(\pi(x))$). However, when $\pi(x)$ lies on an edge (or vertex) of the mesh, it is simultaneously associated with two (or three) triangles. In this case, we select the normal vector of the triangle with the smallest index as the normal direction for $\pi(x)$. Here, we leverage the property of the `torch.argmin` function.

### D.4 High-dimensional special orthogonal group

The manifold $SO(10)$ is defined as $\{Q \in \mathbb{R}^{10 \times 10} | QQ^T = I_{10}, \det(Q) = 1\}$, which represents (a connected component of) the zero-level set of the map $\xi : \mathbb{R}^{100} \to \mathbb{R}^{55}$. The components of $\xi$ correspond to the upper triangular portion of the matrix $QQ^T - I_{10}$. The dataset is constructed as a mixture of 5 wrapped normal distributions. Each wrapped normal distribution is the image (under the exponential map) of a normal distribution defined in the tangent space at a pre-selected center. For more details, refer to [23]. In addition, Figure 3 shows that compared with Iso-DM, the samples from Niso-DM have a better match with the ground truth.

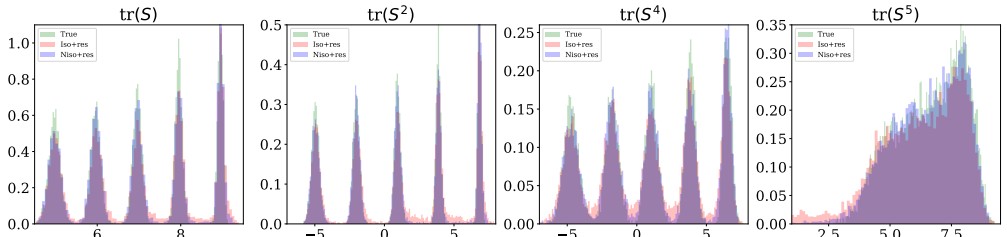

Figure 3: Results for $SO(10)$. Empirical densities of the statistics $\mathrm{tr}(S), \mathrm{tr}(S^2), \mathrm{tr}(S^4)$, and $\mathrm{tr}(S^5)$ for Iso-DM (red), Niso-DM (purple) and the ground truth (green). Samples from Niso-DM (purple) better match the ground truth (green).

### D.5 Alanine dipeptide

To generate the dataset, we follow [23] to obtain the target distribution on the manifold $\mathcal{M} = \{x \in \mathbb{R}^{30} | \phi(x) = -70°\}$. We also ensure that the generated distribution is SE(3)-invariant (i.e., invariant under rotations and translations) by incorporating alignment before feeding the data into the neural network.

## E Ablation Study

Recall that the noise scale $\sigma_t$ is chosen as $\sigma_t = \sigma_{\min}(\sigma_{\max}/\sigma_{\min})^{t/T}$ and $0 < \sigma_{\min} \ll 1$. When $t$ is sufficiently small, $\sigma_t$ approaches $\sigma_{\min}$, and the singularity of the score function primarily depends on $\sigma_{\min}$. As $\sigma_{\min}$ decreases, the singularity becomes pronounced.

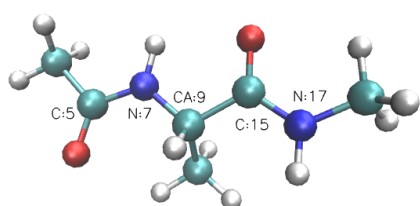

For Iso-DM, there exists a bias-variance trade-off in the choice of $\sigma_{\min}$. A very small $\sigma_{\min}$ reduces bias but increases variance, resulting in a larger overall error. In this case, $\nabla_x \log p_{\sigma_{\min}}(x)$ approximates $\nabla_x \log p_0(x)$ well; however, severe multiscale discrepancies between its tangential and normal components hinder learning. Conversely, a very large $\sigma_{\min}$ increases bias and reduces variance, yet the overall error still increases. In this scenario, $\nabla_x \log p_{\sigma_{\min}}(x)$ fails to serve as a good approximation of

Figure 4: Illustration of the Alanine dipeptide system: The dihedral angles $\phi$ and $\psi$ are defined by atoms whose indices are $5, 7, 9, 15$ and $7, 9, 15, 17$, respectively. This figure is from [23].

$\nabla_x \log p_0(x)$, although the singularity issue is less pronounced. Figure 5a illustrates this phenomenon, where the red line represents the error for Iso-DM.

For Niso-DM, the singularity is governed by the additional noise scale $c_{\text{niso}}$, rather than $\sigma_{\min}$. As a result, our method exhibits greater robustness as $\sigma_{\min}$ decreases. The blue line in Figure 5a demonstrates the stability of the error for Niso-DM under varying $\sigma_{\min}$.

Figures 5b and 5c show the impact of $c_{\text{niso}}$ in Niso-DM and $c_{\text{tango}}$ in Tango-DM on the error. The selection of these hyperparameters also involves a bias-variance trade-off.

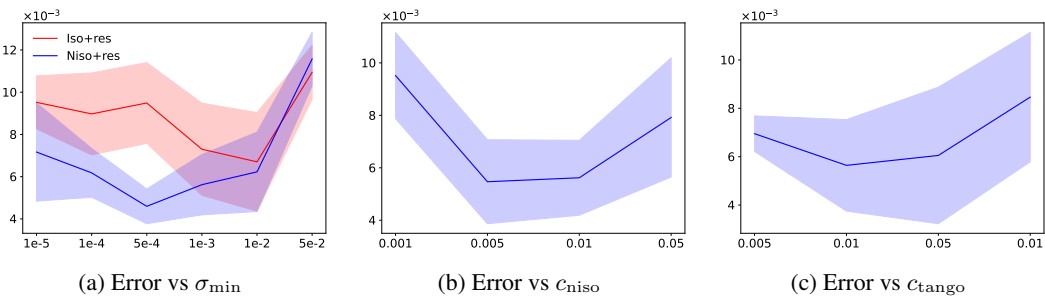

(a) Error vs $\sigma_{\min}$       (b) Error vs $c_{\text{niso}}$       (c) Error vs $c_{\text{tango}}$

Figure 5: Ablation Studies for $\mathrm{SO}(10)$: The solid line denotes the mean, while the shaded area indicates the standard deviation. (a) The error of the distribution generated by the Reverse SDE algorithm under the Iso-DM (red) and Niso-DM (blue) methods with varying $\sigma_{\min}$. (b) The impact of $c_{\text{niso}}$ on the error with $\sigma_{\min} = 0.001$. (c) The impact of $c_{\text{tango}}$ on the error with $\sigma_{\min} = 0.001$.

In Tables 6-9, we present the numerical results of ablation studies on $\sigma_{\min}$, $c_{\text{niso}}$, and $c_{\text{tango}}$, for the $\mathrm{SO}(10)$ experiment. We report computational errors under different parameter settings to assess the sensitivity of our method to hyperparameter choices. The results demonstrate that reductions in hyperparameter values may not substantially affect the performance of our method.

Table 6: The impact of $\sigma_{\min}$ on the error under the Niso algorithm.

| $\sigma_{\min}$ | 1e-5 | 1e-4 | 5e-4 | 1e-3 | 1e-2 |
|---|---|---|---|---|---|
| Iso | 9.52e-3$_{\pm1.25\text{e-}3}$ | 8.97e-3$_{\pm1.94\text{e-}3}$ | 9.49e-3$_{\pm1.91\text{e-}3}$ | 7.30e-3$_{\pm2.18\text{e-}3}$ | 6.70e-3$_{\pm2.33\text{e-}3}$ |
| Niso | 7.17e-3$_{\pm2.33\text{e-}3}$ | 6.18e-3$_{\pm1.16\text{e-}3}$ | 4.60e-3$_{\pm8.24\text{e-}4}$ | 5.62e-3$_{\pm1.43\text{e-}3}$ | 6.23e-3$_{\pm1.88\text{e-}3}$ |

Table 7: The impact of $c_{\text{niso}}$ on the error under the Niso algorithm.

| $c_{\text{niso}}$ | 1e-3 | 5e-3 | 1e-2 | 5e-2 |
|---|---|---|---|---|
| Niso | 9.52e-3$_{\pm1.64\text{e-}3}$ | 5.47e-3$_{\pm1.60\text{e-}3}$ | 5.62e-3$_{\pm1.43\text{e-}3}$ | 7.92e-3$_{\pm2.27\text{e-}3}$ |

# F   Discussion

## F.1   Computational cost of Tango-DM

While Tango-DM tends to be slower due to the annealing sampler, the runtime is primarily determined by the number of steps of the inner Langevin dynamics. For example, in the toy experiment R2inR3, the sampling time for Reverse SDE is 1.68 seconds, while for Annealing SDE, the sampling time increases to 5.62 seconds with an inner step of 5 and to 9.40 seconds with an inner step of 10. The first phase of Annealing SDE (see Algorithm 3) requires only 0.82 seconds. Overall, the computational cost of Annealing SDE is approximately 3–6 times higher than that of Reverse SDE. These measurements, conducted on a standard laptop, highlight the relative computational expense.

## F.2   Learned manifold case

Unlike previous works on distributions on a known manifold, our method avoids relying on geometric information like geodesics or heat kernels, greatly reducing computational complexity. However, compared to approaches under the manifold hypothesis, our method still requires knowledge of the manifold's definition, including projection operators. In fields like image and language processing, the manifold structure is often assumed but not explicitly known.

Table 8: The impact of $\sigma_{\min}$ on the error under the Tango algorithm.

| $\sigma_{\min}$ | 1e-5 | 1e-4 | 5e-4 | 1e-3 | 1e-2 |
|---|---|---|---|---|---|
| Niso | 8.60e-3±1.05e-3 | 8.12e-3±2.33e-3 | 6.42e-3±1.97e-3 | 6.05e-3±2.82e-3 | 4.11e-2±1.82e-3 |

Table 9: The impact of $c_{\text{tango}}$ on the error under the Tango algorithm.

| $c_{\text{tango}}$ | 5e-3 | 1e-2 | 5e-2 | 1e-1 |
|---|---|---|---|---|
| Tango | 6.95e-3±7.37e-4 | 5.64e-3±1.89e-3 | 6.05e-3±2.82e-3 | 8.46e-3±2.67e-3 |

Next, we discuss how to extend our method to learned manifolds, using the AutoEncoder as an example. Specifically, we first train an Encoder $\phi_{\theta_1} : \mathcal{M} \to \mathbb{R}^d$ and a Decoder $\psi_{\theta_2} : \mathbb{R}^d \to \mathcal{M}$, which provide a parameterized representation of the manifold. The subspace spanned by $\nabla \psi_{\theta_2}(\phi_{\theta_1}(x))$ corresponds to the tangent space $T_x\mathcal{M}$ at point $x$ on the manifold. Once the basis of the tangent space is obtained, the score function can be further decomposed into its tangential and normal components, which enables the implementation of our proposed methods. Intuitively, the success of this approach hinges on the Autoencoder's ability to accurately capture the underlying manifold structure. In this work, we leave this approach as a direction for future research.

### F.3 Assumptions in Theorem 3.1 and Theorem 4.1

The conclusions in Theorem 3.1 and Theorem 4.1 hold pointwise on the manifold, so in the proof, we only require local boundedness. Specifically, the following assumption is sufficient:

- For any $x \in \mathcal{M}$, there exists $\delta > 0$, such that $\sup_{y \in \mathcal{M}_x^\delta} \max_{1 \leq i,j,j' \leq n} \left| \frac{\partial P_{ij}}{\partial y_{j'}}(y) \right| < +\infty$, where $\mathcal{M}_x^\delta = \{\arg\min_{x^* \in \mathcal{M}} |\tilde{x} - x^*|^2 \mid |\tilde{x} - x| < \delta\}$.

To make the theorem statement more concise, we adopted a stronger assumption:

- $\sup_{x \in \mathcal{M}} \max_{1 \leq i,j,j' \leq n} \left| \frac{\partial P_{ij}}{\partial x_{j'}}(x) \right| < +\infty$.

For compact manifolds, the uniform boundedness assumptions always hold, which implies that the local boundedness assumptions hold. Similarly, the boundedness assumption in Theorem 4.1 can also be relaxed to local boundedness.

## G   Impact statement

This paper contributes to the advancement of generative models for data with manifold structures, providing a deeper understanding of the singularity of the score function. Specifically, we identify the scale discrepancies between the tangential and normal components of the score function, which sheds light on key challenges in modeling data on manifolds. We believe that our work bridges the gap between generative models specifically designed for manifolds and those aimed at handling data under manifold assumptions. While this study may have broader societal implications, none require particular emphasis at this stage.

