# OpenReview forum: "Improving the Euclidean Diffusion Generation of Manifold Data by Mitigating Score Function Singularity"
_NeurIPS.cc/2025/Conference — NeurIPS 2025 poster_

### Official Review · Reviewer_TsMq · 2025-06-11

**Clarity:** 3
**Significance:** 2
**Originality:** 2
**Rating:** 4
**Confidence:** 4

**Summary:**

The paper identifies a potential issue with generative diffusion with data supported on manifolds, the (well known)  fact that the orthogonal component of the score diverges as 1/sigma for t-> 0 while its tangential component approaches a constant. Based on this observation, the authors suggest a series of methods to stabilize both training and sampling essentially by separating the treatment of the two components. Their training method trains the orthogonal and tangential component separately and it can only be directly used when the m manifold structure is known. The increase in performance for toy models is small when compared with the standard and widely used rescaled approach, where the divergent term 1/sigma is factored out from the network.

**Questions:**

Can your method be used in real data where the manifold structure is not know? If not, is it possible to modify the approach so that it can be used for images or other kinds of low-dimensional data?

**Ethical Concerns:**

["NO or VERY MINOR ethics concerns only"]

**Final Justification:**

I do think that the revised manuscript will be useful to the community as it formally characterize an important phenomenon and suggests an interesting mitigation technique
The authors should include the extended coverage of the literature in the revised manuscript.

**Limitations:**

I would have apprecoiated to see a proper discussion of the limitations in the main text.

**Paper Formatting Concerns:**

Everything looks fine

**Quality:**

3

**Strengths And Weaknesses:**

This paper seems to imply that standard generative diffusion cannot deal well with manifold data. I think that this assertion is wrong as standard techniques can perform very well on data supported on manifolds and in fact they shine in this regime since alternative likelihood based methods are destabilized by the singularity of the loss function (see manifold overfitting [1]). The divergence of the score has already been studied extensively, see for example in [2,3,4,5,6]. The analysis in [3] is particularly interesting since it suggests that the ‘irrelevance’ of the tangential component of the score near zero is a feature, not a bug as in that phase (manifold consolidation) the role of the score is to project the samples on manifold since the intrinsic distribution of the manifold has been determined at earlier times (manifold coverage phase). This interpretation explains why,in general, diffusion models do not suffer from manifold overfitting.

So said, I do think that this topic definitely deserves further analysis and I do think that the method proposed by the authors can have value in specific scenarios where the correct estimation of the tangential components at times close to zero is vital. The experimental results are interesting and I can see the proposed methods to have some use in scientific applications.

A major problem with the paper is that it ignores a very large body of existing literature on the subject. It is vital to properly engage with previous work and it seems that the authors missed some very relevant papers. Here is a selected list:

[1] Stanczuk, Jan, et al. "Your diffusion model secretly knows the dimension of the data manifold." arXiv preprint arXiv:2212.12611 (2022).
[2] Ventura, Enrico, et al. "Manifolds, Random Matrices and Spectral Gaps: The geometric phases of generative diffusion." ICLR (2025)
[3] Kadkhodaie, Zahra, et al. "Generalization in diffusion models arises from geometry-adaptive harmonic representations." ICLR (2024)
[4] Wang, Peng, et al. "Diffusion models learn low-dimensional distributions via subspace clustering." arXiv preprint arXiv:2409.02426 (2024).
[5] Achilli, Beatrice, et al. "Memorization and Generalization in Generative Diffusion under the Manifold Hypothesis." arXiv preprint arXiv:2502.09578 (2025).
[6] George, Anand Jerry, Rodrigo Veiga, and Nicolas Macris. "Analysis of Diffusion Models for Manifold Data." arXiv preprint arXiv:2502.04339 (2025).

I highly recommend the authors to incorporate this literature in their paper.

[edit] I in creased my score due to the work of the authors on incorporating a discussion  of the relevant literature.

---

> ### Author Rebuttal · Authors · 2025-07-31
>
> We are grateful to the reviewer for the careful review and constructive comments.
>
> ## Summary
>
> First, we would like to kindly point out that the reviewer’s summary of our work is somewhat unfair and inaccurate. The reviewer claims that "the orthogonal component of the score diverges as $1/\sigma$ for $t \to 0$ while its tangential component approaches a constant" is a "well-known fact." This statement is unfair. The truly well-known fact is the divergence of the score and manifold overfitting phenomenon when the data is supported on manifolds, rather than this detailed statement about the behavior of its orthogonal and tangential components we provided. Although some earlier works such as **[1], [3] implicitly hinted** at the scale differences in spirit, the observation that "the orthogonal component of the score diverges as $1/\sigma$ for $t \to 0$ while its tangential component is of order 1" is **explicitly and mathematically formulated by us for the first time in the known manifold setup, accompanied by rigorous mathematical proofs**. Reviewers 42Gv and vZYH have also acknowledged that our result on the scale discrepancy is the first analysis in this field.
>
> Regarding the experimental evaluation, the reviewer stated that "the increase in performance for toy models is *small*." We respectfully disagree with this assessment. We believe these experiments effectively demonstrate the validity of our proposed method and confirm the correctness of our theoretical analysis.
>
> ## Strengths And Weaknesses
>
>
> We have **never claimed that "standard generative diffusion cannot deal well with manifold data"**, which seems to be the viewpoint that the reviewer is attempting to refute. **Our actual argument is that mitigating the scale discrepancy of the score function can improve the performance of diffusion models, while it is true that diffusion models outperform likelihood-based methods.**
>
> The statement that "the 'irrelevance' of the tangential component of the score near zero is a feature, not a bug, as in that phase (manifold consolidation) the role of the score is to project the samples onto the manifold since the intrinsic distribution of the manifold has been determined at earlier times (manifold coverage phase)" **can, at best, explain why diffusion models can work in manifold settings or outperform likelihood-based methods in such scenarios.** However, **the precision of the intrinsic/internal distribution on the manifold determined at earlier times (manifold coverage phase) may not be sufficient**. The tangential component of the score near zero time (manifold consolidation phase) **allows for a more refined characterization of the intrinsic distribution of the manifold** and consequently is not an "irrelevant" part. This is one of the motivations behind our proposed methods, i.e., Niso and Tango. We believe that this statement may not be sufficient to serve as evidence against the arguments in our paper.
>
> We believe that the reviewer has fully understood our motivation, as evidenced by their statement: "… have value in specific scenarios where the correct estimation of the tangential components at times close to zero is vital." However, we would like to argue that this is not limited to specific scenarios but rather represents a general setting.
>
> In fact, if one truly aims to accurately compute the tangential component at times close to zero, we believe our methods effectively address this problem and lay the foundation for future studies. **As shown in Figure 1 in our paper, our method significantly reduces the error of the tangential component at times close to zero**, thereby improving overall accuracy.
>
> **We apologize for overlooking relevant literature and thank the reviewer for highlighting these important references**. We greatly appreciate their insightful feedback and thorough knowledge of the field. Specifically, we find reference [3] interesting because its Jacobian eigendecomposition of the denoiser mapping also implicitly reflects scale differences, which aligns with the statements in our paper.
>
> We will incorporate the following discussion of these papers in the related work section in the revised version.
>
>
> > The work [1] by Stanczuk et al. essentially uses the score discrepancy to identify the intrinsic dimension of the manifold. Kadkhodaie et al. [3] describe denoising in terms of basis functions, which implies the scale discrepancy we discussed. In particular, the Jacobian eigendecomposition of the denoiser mapping (Eq. (12)) implicitly captures these scale differences, aligning with Eq. (10) in our paper. However, our work explicitly reveals this phenomenon in manifold settings using more formal mathematical formulations and provides solutions to address it. References [4] [5] [6] study the fundamental principles of learning manifold-structured data distributions through Gaussian Mixture Models, Hidden Manifold Models, and Generalized Linear Models. References [1] and [2] also investigate the intrinsic geometry of generative diffusion models by studying the eigendecomposition of the Jacobian of the score functions. Moreover, we believe that our proposed methods can further improve the accuracy of diffusion models during the manifold consolidation phase as described in [2].
>
>
> ## Questions: Real Data Case
>
> Our current algorithm requires explicit knowledge of the manifold structure, which indeed limits its applicability to image or language data, where the manifold is often assumed but not explicitly known.
>
> One possible approach to extending our algorithm is to train an Autoencoder to learn the manifold structure. Specifically, this involves learning an Encoder $\phi_{\theta_1}: \mathcal{M} \to \mathbb{R}^d$ and a Decoder $\psi_{\theta_2}: \mathbb{R}^d \to \mathcal{M}$, which serve as a parameterized representation of the manifold. The subspace spanned by $\nabla\psi_{\theta_2}(\phi_{\theta_1}(x))$ corresponds to the tangent space at point $x$ on the manifold. After obtaining a basis of the tangent space, we can further decompose the score function into its tangential and normal components. In principle, our method can also work on manifolds learned via an Autoencoder. However, in this work, we do not plan to implement this approach and instead leave it for future research.
>
> We will add a discussion on this limitation in the main text (see next section).
>
> ## Limitations
>
> We will address this limitation explicitly in the main text. As follows:
>
> > Unlike previous works on distributions on a known manifold, our method avoids relying on geometric information like geodesics or heat kernels, greatly reducing computational complexity. However, compared to approaches under the manifold hypothesis, our method still requires knowledge of the manifold’s definition, including projection operators. In fields like image and language processing, the manifold structure is often assumed but not explicitly known. Future work will focus on extending our approach to scenarios where the manifold is undefined or implicit.
>
> ## Final Words
>
> Overall, our work provides a mathematically explicit analysis of the singular structure of the score function along its tangential and normal components, even though the divergence of the score function has already been widely studied. Furthermore, we propose two methods to improve the performance of diffusion models by mitigating the scale discrepancy of the score function. If one aims to improve accuracy by precisely computing the tangential component near $t=0$, we believe our approach captures the essence of the problem and has the potential to inspire future research.
>
> While we acknowledge that some relevant papers were overlooked in the current version, we believe the novelty of our approach is still evident. We will incorporate discussions of the related work in the revised version. Once again, we sincerely thank the reviewer for their constructive feedback.

---

> > ### Comment · Reviewer_TsMq · 2025-08-03
> >
> > Dear authors,
> >
> > Thank you for the reply. I do appreciate that this works provide a detailed and rigorous characterization of the phenomenon, but it remains the fact that this result has been common knowledge among my collaborators and me for quite a long time and it is implied in several prior works such as [1] for example. My point is that the results are not particularly original, but I acknowledge that the analysis is well executed and possibly useful.
> >
> > I think that in this case it is crucial to properly discuss the literature I mentioned and possibly other works that I might have missed, I do think that the lack of the literature coverage is one of the main weaknesses of the paper. Could you provide the detailed revised Related Work paragraph here in the response to this post? This is crucial for me to increase my score.

---

> > > ### Author Response · Authors · 2025-08-04
> > > **Revised Related Work Section Part 1**
> > >
> > > We sincerely appreciate the reviewers' feedback and their recognition of the strengths of our work to some extent. We once again thank the reviewer for pointing out these highly relevant and interesting works.
> > >
> > > We plan to revise the related work section as follows, including two new paragraphs: **Scale discrepancy of the score function** and **Additional noise along the normal direction**, the latter of which is suggested by Reviewer 42Gv. Should there be additional suggestions, we remain open to refining our related work section.
> > >
> > > > Research on manifold-related diffusion models can be broadly categorized into two directions: (1) diffusion models for distributions under the manifold hypothesis, where the underlying manifold is unknown, and (2) diffusion models for distributions on a known manifold, which is the focus of this work. For the former, we discuss the singularity and the scale discrepancy of score functions mentioned in the related work; for the latter, we explore several studies based on known manifolds.
> > > >
> > > > **Singularity of the score function.** Several studies on diffusion models under the manifold hypothesis have highlighted the divergence of the score function as $t \to 0$. Beyond empirical observations, recent studies have provided mathematical analysis of diffusion models based on the VPSDE. For instance, Chen et al. present a mathematical analysis under the assumption that each data point lies on a hyperplane. Pidstrigach et al. show that the norm of the score function satisfies $ \mathbb{E}\|\nabla_x \log p_{t}(x))\| \gtrsim 1/\sqrt{t} $. Furthermore, Lu et al. rigorously prove that this singularity follows a $ 1/t $ scaling in a strong pointwise sense under the VPSDE, which is similar to our results in Eq (10). Further discussion can be found in Loaiza-Ganem et al..
> > > >
> > > > To mitigate this singularity issue, a number of works (e.g. Song et al.) assume that the initial time is at $t=\epsilon>0$, instead of $t = 0$. The study by Dockhorn et al. introduces a diffusion model on the product space of position and velocity, employing hybrid score matching to circumvent the singularity. Furthermore, in Song et al., the score function is parameterized by $s_\theta(x, t) = \hat{s}_\theta(x)/\sigma_t$, where $\hat{s}_\theta(x)$ is a neural network.
> > > >
> > > > **Scale discrepancy of the score function.** The work [1] uses the singular value decomposition of the score matrix to identify the intrinsic dimension of the manifold, which essentially utilizes the discrepancy of the score function. References [2] [3] [4] [8] investigate the fundamental principles of diffusion models under the manifold hypothesis, by studying the eigen-decomposition of the Jacobian of the score functions. This analysis implicitly relates to the score discrepancy discussed in our work. In particular, the decomposition of the denoiser mapping (Eq (12)) in [3], which involves a geometry-adaptive harmonic basis, also implies this discrepancy, aligning with Eq (10) in our paper. Papers [5] and [6] study the Memorization and Generalization in Generative Diffusion under the manifold hypothesis through Hidden Manifold Models and Generalized Linear Models. Besides, [9] suggests that a conservative diffusion is guaranteed to yield the correct conclusions when analyzing local features of the data manifold.
> > > >
> > > > While previous works have implicitly touched upon the scale differences in manifold settings, our work explicitly formalizes this phenomenon with rigorous mathematical formulations and introduces novel solutions to address it. Furthermore, we believe our methods can enhance the accuracy of diffusion models during the manifold consolidation phase (as described in [2]) or the collapse transition times (as mentioned in [5] [6]), by refining the computation of the tangential component of the score function.
> > > >
> > > > **Additional noise along the normal direction.** Paper [7] proposes adding noise along the normal direction to "inflate" manifolds under the setting of normalizing flows, which is similar to our Niso method. Theorem 4 and Proposition 7 in [7] analyze the relationship between the perturbed distribution and the original distribution on the manifold, which aligns with the conclusions in our equations (12) and (18).
> > > >
> > > > On the other hand, our method differs from [7] in several key aspects. In [7], the added noise has a constant magnitude, requiring additional assumptions (e.g., Q-normally reachable) to prevent interference between noise at different points. In contrast, our analysis (Theorem 3.1) permits the noise magnitude to diminish toward zero, relying on weaker assumptions. Additionally, [7] applies noise inflation prior to training the normalizing flow, whereas our method integrates noise addition directly into the diffusion process, accompanied by the noise dynamics of diffusion models. Finally, while the analysis in [7] focuses on the perturbed density function, our work studies the asymptotic behavior of the score function.

---

> > > > ### Author Response · Authors · 2025-08-04
> > > > **Revised Related Work Section Part 2**
> > > >
> > > > > **Diffusion models on manifolds.** Riemannian Score-based Generative Models extend SDE-based diffusion models to manifold settings by estimating the transition kernels using either the Riemannian logarithmic map or the eigenpairs of the Laplacian–Beltrami operator. Riemannian Diffusion Models employ a variational diffusion framework for Riemannian manifolds and, similar to our approach, consider submanifolds embedded in Euclidean space. Additionally, Trivialized Diffusion Models adapt diffusion models from Euclidean spaces to Lie groups.
> > > >
> > > >
> > > > - [1] Stanczuk, Jan, et al. "Your diffusion model secretly knows the dimension of the data manifold." arXiv preprint arXiv:2212.12611 (2022). Another name: Diffusion Models Encode the Intrinsic Dimension of Data Manifolds, ICML (2024).
> > > > - [2] Ventura, Enrico, et al. "Manifolds, Random Matrices and Spectral Gaps: The geometric phases of generative diffusion." ICLR (2025).
> > > > - [3] Kadkhodaie, Zahra, et al. "Generalization in diffusion models arises from geometry-adaptive harmonic representations." ICLR (2024).
> > > > - [4] Wang, Peng, et al. "Diffusion models learn low-dimensional distributions via subspace clustering." arXiv preprint arXiv:2409.02426 (2024).
> > > > - [5] Achilli, Beatrice, et al. "Memorization and Generalization in Generative Diffusion under the Manifold Hypothesis." arXiv preprint arXiv:2502.09578 (2025).
> > > > - [6] George, Anand Jerry, Rodrigo Veiga, and Nicolas Macris. "Analysis of Diffusion Models for Manifold Data." arXiv preprint arXiv:2502.04339 (2025).
> > > > - [7] Christian Horvat, Jean-Pascal Pfister. "Density estimation on low-dimensional manifolds: an inflation-deflation approach." JMLR (2023).
> > > > - [8] Wenliang, Li Kevin, and Ben Moran. "Score-based generative models learn manifold-like structures with constrained mixing." NeurIPS 2022 Workshop on Score-Based Methods.
> > > > - [9] Christian Horvat, Jean-Pascal Pfister. "On gauge freedom, conservativity and intrinsic dimensionality estimation in diffusion models." ICLR (2024).

---

> > > > ### Comment · Reviewer_TsMq · 2025-08-05
> > > >
> > > > I wish to thank the authors for their work on integrating the relevant literature into their manuscript, which I think makes this work substantially stronger.
> > > > All in all, I do think that this work is valuable and has several novel elements such as the mitigation technique, and I will therefore raise my score to 4 and recommend acceptance, conditional on the inclusion of the extended literature coverage in the camera ready version.

---

> > > > > ### Author Response · Authors · 2025-08-05
> > > > >
> > > > > We sincerely thank the reviewer for their valuable suggestions and the time and effort they dedicated to our work.
> > > > >
> > > > > Best regards,
> > > > >
> > > > > The Authors

---

### Official Review · Reviewer_vZYH · 2025-06-17

**Clarity:** 3
**Significance:** 3
**Originality:** 2
**Rating:** 5
**Confidence:** 4

**Summary:**

The paper identifies, for data on a low-D manifold, different magnitude of the score matching loss tangential and normal to the local manifold. The authors thus proposed two modifications to the score function for know manifolds. Experiments show that the modifications, usually when using the "epsilon" equivalent of the score matching loss, gives improvements on toy and real datasets.

**Questions:**

1. I'm curious about the sensitivity of hyperparameters mentioned before.
2. Can you combine niso and tango with some weighting to get even better performance?

**Ethical Concerns:**

["NO or VERY MINOR ethics concerns only"]

**Final Justification:**

The authors provided new / pointed to existing experimental results that resolved my concerns. The related work are more thoroughly acknolwedged. Limitations more explicitly discussed.

**Limitations:**

I think the limitation are not clearly described, but only touched on during writing. Please at least mention that the method requires knowing the manifold, which is not always the case in machine learning applications, e.g. on images.

**Quality:**

4

**Strengths And Weaknesses:**

# Strengths
1. Many real-world data distributions fall under the manifold hypotheses. This paper addresses the score discrepancies within and normal to the manifold. This result is not entirely new, but as far as I know, the analyses the authors provide are first in the field.
2. Using knowledge of the manifold, the authors proposed two intuitive methods to address the problem. These methods are clearly described with pros and cons discussed.
3. The paper is written clearly with proper use of mathematical notations, and I find it easy to learn and absorb the non-trivial contributions and ideas.
4. The authors tested the proposed remedy on a wide range of datasets and problems, with promising results.

# Weaknesses:

1. Using knowledge of the manifold is a big assumption, and is suitable only for low dimensional problems.
2. A lack of baseline methods that also use knowledge of the given manifold. Are the authors propose to use manifold information? Or to use manifold information in the particular ways the authors propose? If the former, then this is somewhat trivial; if the latter, then we need some other baselines. A easy baseline is to improve the iso with manifold information: project the samples to or towards the manifold with some parameters. This should give us a the performance of a naive diffusion + manifold method.
3. A lack of Ablations and sensitivity analyses: for tango, how important is the choice of $\sigma_t$ that we train only the tangent score? For niso, how sensitive is the performance to $\alpha_t$?
4. The authors cited a few related work, but I think it's missing important conplementary / congruent prior work on showing the discrepancy of within- and out-of-manifold scores, though with less mathematical formalism and without using a given manifold function $\xi$.
    1. [*] uses a heuristic argument to show that the score perpenticular to noisy data of images is mostly Gaussian-like, whereas the score within the manifold as additional properties (no directly relevant for the current submission)
    2. [**] Essentially uses the score descrepancy to identify the manifold.
    3. [***] Describes denoising in terms of bases functions. In particular, Eqn (12) also embodies the scale discrepancy the authors of the current paper describe here.
All of these methods do not assume a given manifold, so it would be great to add these to the Related work, part 1 where you mention other methods that do not rely on a known manifold.

[*] Wenliang, Li Kevin, and Ben Moran. "Score-based generative models learn manifold-like structures with constrained mixing." NeurIPS 2022 Workshop on Score-Based Methods.

[**] Stanczuk, Jan, et al. "Your diffusion model secretly knows the dimension of the data manifold." arXiv preprint arXiv:2212.12611 (2022).

[***] Kadkhodaie, Zahra, et al. "Generalization in diffusion models arises from geometry-adaptive harmonic representations." The Twelfth International Conference on Learning Representations.

---

> ### Author Rebuttal · Authors · 2025-07-31
>
> We sincerely thank the reviewer for the thoughtful comments and valuable recommendations.
>
> ## Strengths
>
> We thank the reviewer for their positive evaluation of our work. By addressing the score discrepancies within and normal to the manifold, our analyses provide a novel contribution to the field. Leveraging manifold knowledge, we proposed two intuitive methods to tackle this problem effectively.
>
> ## Weaknesses
>
> ### W1: Manifold Assumption
>
> Our current algorithm requires explicit knowledge of the manifold structure, which indeed limits its applicability to image or language data, where the manifold is often assumed but not explicitly known.
>
> One possible approach to extending our algorithm is to train an Autoencoder to learn the manifold structure at first. Specifically, this involves learning an Encoder $\phi_{\theta_1}: \mathcal{M} \to \mathbb{R}^d$ and a Decoder $\psi_{\theta_2}: \mathbb{R}^d \to \mathcal{M}$, which serve as a parameterized representation of the manifold. The subspace spanned by $\nabla\psi_{\theta_2}(\phi_{\theta_1}(x))$ corresponds to the tangent space at point $x$ on the manifold. After obtaining a basis of the tangent space, we can further decompose the score function into its tangential and normal components. In principle, our method can also work on manifolds learned via an Autoencoder. However, in this work, we do not plan to implement this approach and instead leave it for future research.
>
>
> ### W2: Baseline
>
> Yes, we aim to use manifold information in these particular ways. The reviewer's suggestion is excellent and precisely reflects our existing approach. The results of the baselines (Iso or Iso+res) shown in the paper are conducted exactly as suggested by the reviewer. Specifically, we use the Reverse SDE sampling algorithm, which includes a projection step as the final step (see lines 205-207 in the main text). In other words, the Iso and Iso+res methods in our paper have already been combined with manifold information. We will make this information more explicit in the revised version of the paper.
>
>
> Besides, we have conducted Riemannian diffusion models (RDM, [*]) on our experiments for comparison. This algorithm, based on sliced score matching and Hutchinson estimation, is applicable to general known manifolds. Therefore, we use RDM as a baseline and conduct experiments on Mesh data and SO(10).
>
> Table 1 presents the comparison between the vanilla algorithm (with a final projection step), our method (using Niso+res with Reverse SDE), and RDM (baseline). Our method consistently outperforms the baseline. Specifically, in the SO(10) case, our approach outperforms RDM, while the vanilla algorithm does not. For Mesh data, RDM does not perform well, which can be explained by the fact that RDM relies on defining a diffusion process on the manifold to achieve a uniform distribution. Due to the geometric structure of the manifold, such as the narrow neck of "Spot the Cow", achieving a uniform distribution via SDEs on manifolds proves challenging.
>
> Table 1: Computational errors for Mesh data and SO(10). The results are reported as the mean and standard deviation over five independent runs.
>
> | Method   | Bunny           | Spot            | SO(10)          |
> |----------|------------------|-----------------|-----------------|
> | RDM      | 7.00e-1 (3.85e-3) | 7.22e-1 (3.32e-3) | 7.14e-3 (9.09e-4) |
> | Vanilla  | 3.52e-1 (2.17e-3) | 3.30e-1 (2.08e-3) | 9.49e-3 (1.91e-3) |
> | Ours     | **3.48e-1** (8.27e-4) | **3.28e-1** (6.19e-4) | **4.60e-3** (8.24e-4) |
>
>
> [*] Huang, Chin-Wei, et al. "Riemannian diffusion models." Advances in Neural Information Processing Systems 35 (2022): 2750-2761.
>
>
> ### W3: Ablation Study
>
> Following the reviewer’s suggestion, we have added ablation studies to evaluate the sensitivity of our method to parameters. In our algorithm, $\sigma_{\rm min}$, $c_{\rm niso}$, and $c_{\rm tango}$ are relatively important hyperparameters. Under the $SO(10)$ experiment, we present the computational errors for different parameter choices. Some of the results below are already contained in Fig. 3 in Appendix E of the paper.
>
> Table 2: The impact of $\sigma_{\rm min}$ on the error under the Niso algorithm.
> | $\sigma_{\rm min}$ | 1e-5             | 1e-4             | 5e-4             | 1e-3             | 1e-2             |
> | ------------------ | ---------------- | ---------------- | ---------------- | ---------------- | ---------------- |
> | Iso                | 9.52e-3(1.25e-3) | 8.97e-3(1.94e-3) | 9.49e-3(1.91e-3) | 7.30e-3(2.18e-3) | 6.70e-3(2.33e-3) |
> | Niso               | 7.17e-3(2.33e-3) | 6.18e-3(1.16e-3) | 4.60e-3(8.24e-4) | 5.62e-3(1.43e-3) | 6.23e-3(1.88e-3) |
>
>
> Table 3: The impact of $c_{\rm niso}$ on the error under the Niso algorithm.
> | $c_{\rm niso}$ | 1e-3             | 5e-3             | 1e-2             | 5e-2             |
> | -------------- | ---------------- | ---------------- | ---------------- | ---------------- |
> | Niso           | 9.52e-3(1.64e-3) | 5.47e-3(1.60e-3) | 5.62e-3(1.43e-3) | 7.92e-3(2.27e-3) |
>
>
> Table 4: The impact of $\sigma_{\rm min}$ on the error under the Tango algorithm.
> | $\sigma_{\rm min}$ | 1e-5             | 1e-4             | 5e-4             | 1e-3             | 1e-2             |
> | ------------------ | ---------------- | ---------------- | ---------------- | ---------------- | ---------------- |
> | Niso               | 8.60e-3(1.05e-3) | 8.12e-3(2.33e-3) | 6.42e-3(1.97e-3) | 6.05e-3(2.82e-3) | 4.11e-2(1.82e-3) |
>
>
> Table 5: The impact of $c_{\rm niso}$ on the error under the Tango algorithm.
> | $c_{\rm tango}$ | 5e-3             | 1e-2             | 5e-2             | 1e-1             |
> | --------------- | ---------------- | ---------------- | ---------------- | ---------------- |
> | Tango           | 6.95e-3(7.37e-4) | 5.64e-3(1.89e-3) | 6.05e-3(2.82e-3) | 8.46e-3(2.67e-3) |
>
>
> These results indicate that reducing the hyperparameters does not significantly impact the performance of our method. Moreover, with different $\sigma_{\rm min}$ (see Table 1), our algorithm consistently outperforms the vanilla method (iso+res). We will include all these numerical results in our paper.
>
>
> ### W4: Citation
> We appreciate the reviewer’s suggestions and will incorporate discussions of these three references into Part 1 of the Related Work section. Specifically, we will include the following points:
>
> > Wenliang et al. show that the score perpendicular to noisy data of images is mostly Gaussian-like, whereas the score within the manifold has additional properties. The work by Stanczuk et al. essentially uses the score discrepancy to identify the intrinsic dimension of the manifold. Kadkhodaie et al. describe denoising in terms of basis functions, which implies the scale discrepancy we discussed. In particular, the Jacobian eigendecomposition of the denoiser mapping (Eq. (12)) implicitly captures these scale differences, aligning with Eq. (10) in our paper. However, our work explicitly reveals this phenomenon in manifold settings using more formal mathematical formulations and provides solutions to address it.
>
>
> ## Questions
>
> ### Q1: Hyperparameters
>
> See the Weaknesses section (W3).
>
> ### Q2: Combination of NISO and Tango
>
> NISO and Tango handle the normal direction in fundamentally different ways: NISO seeks to relax the scale of the normal component, whereas Tango directly discards it. In the paper, NISO and Tango are introduced as two independent algorithms. It seems infeasible to effectively combine them as a new approach.
>
>
> ## Limitations
> We will address this limitation explicitly in the main text. As follows:
>
> > Unlike previous works on distributions on a known manifold, our method avoids relying on geometric information like geodesics or heat kernels, greatly reducing computational complexity. However, compared to approaches under the manifold hypothesis, our method still requires knowledge of the manifold’s definition, including projection operators. In fields like image and language processing, the manifold structure is often assumed but not explicitly known. Future work will focus on extending our approach to scenarios where the manifold is undefined or implicit.
>
> ## Final Words
>
> Overall, our work provides a mathematically explicit analysis of the singular structure of the score function along its tangential and normal components, even though the divergence of the score function has already been widely studied. Furthermore, we propose two methods to improve the performance of diffusion models by mitigating the scale discrepancy of the score function.
>
> While we acknowledge that some relevant papers were overlooked in the current version, we believe the novelty of our approach is still evident. We will incorporate discussions of the related work in the revised version. Once again, we sincerely thank the reviewer for their constructive feedback.

---

> > ### Author Response · Authors · 2025-08-05
> >
> > Dear Reviewer,
> >
> > We appreciate your valuable feedback and suggestions. We have addressed your comments and incorporated an ablation study, a new baseline, and improvements to the related work section. Specifically, regarding the related work section, please refer to our latest Official Comment addressed to everyone.
> >
> > **If you have any remaining concerns or feedback, we would greatly appreciate it if you could share them.**
> >
> > Thank you for your time and input.
> >
> > Best regards,
> >
> > The Authors

---

> > ### Comment · Reviewer_vZYH · 2025-08-05
> > **Asking for a bit more details.**
> >
> > I thank the reviewers for their detailed response and experiments directly addressing my concerns. I have raised my score and hope to see this paper published. I just have one small question:
> >
> > Does the following idea make sense: train an unconstrained score estimator with a linear weighted combination of the two loss functions? Or even with a schedule that depends on the noise scale or t?

---

> > > ### Author Response · Authors · 2025-08-07
> > > **Linear combination of the two loss functions**
> > >
> > > The reviewer has raised an excellent question. Let us consider the following combined loss function, defined as
> > > $$
> > > l_{\text{combine}}(t; \theta) = l_{\rm Niso}(t; \theta) + \lambda l_{\rm Tango}(t; \theta),
> > > $$
> > > which can also be interpreted as a linear combination of the two methods.
> > >
> > > One way to understand $ l_{\text{combine}}(t; \theta) $ is to treat $ \lambda l_{\rm Tango}(t; \theta) $ as a regularization term for our Niso method. From this perspective, the combined loss function can be seen as an enhanced version of the Niso loss, designed to compute the tangential component of the score more precisely.
> > >
> > > Unfortunately, preliminary numerical experiments indicate that this approach did not significantly improve the performance of the Niso method. Table 1 summarizes the test results on our R2inR3 dataset.
> > >
> > > Table 1: Results for R2inR3 under the combined loss function with different $\lambda$. The first column ($\lambda = 0$) represents our Niso algorithm.
> > > | $\lambda$ | 0.0   |  0.1  | 0.5   | 1.0  | 5.0 |
> > > | -------- | --- | --- | --- | --- |--- |
> > > | Error | 1.45e-4(2.68e-5)  | 1.45e-4(2.72e-5) | **1.44e-4**(2.65e-5) | **1.44e-4**(2.67e-5) | 1.51e-4(2.55e-5) |
> > >
> > > This question remains intriguing and deserves further exploration. For instance, one could make $\lambda$ a time-dependent weight $\lambda(t)$.
> > >
> > > **Finally, we would like to once again thank the reviewer for their valuable suggestion and the time and effort they dedicated to our work.**
> > >
> > > Best regards,
> > >
> > > The Authors

---

### Official Review · Reviewer_3q97 · 2025-06-22

**Clarity:** 3
**Significance:** 3
**Originality:** 3
**Rating:** 4
**Confidence:** 2

**Summary:**

When trying to learn data that lies on a lower dimensional manifold with diffusion models that operate on Euclidean embeddings, the optimal score function exhibits a singularity at small noise scales $\sigma \rightarrow 0$. This work attempts to characterize this singularity and proposes two changes to the loss function to remove it from the training process. The authors first split the score function into a tangential and normal component w.r.t. the manifold and note that around $\sigma \rightarrow 0$ the loss contribution of the former grows as $O(1)$, while the latter as $O(\sigma^{-1})$. Therefore the normal component dominates the optimization process (manifold overfitting) at small noise scales. The work proceeds by proposing two variants of the diffusion loss function to fix this:
1. Niso-DM uses non-isotropic Gaussian noise for small $\sigma$, where the noise in the normal direction is increased, giving direct control over the behaviour of the score function in the normal direction.
2. Tango-DM only learns the tangential components of the score function in the regime of small $\sigma$. Sampling in this regime is achieved by annealing the reverse SDE restricted on the manifold, which circumvents the need to know the score in the normal direction.

The authors show that all presented methods produce the correct on-manifold distributions (after projecting the samples to the manifold if required). Experimentally the authors validate their methods on a GMM on a 2-d plane embedded in 3-d, two synthetic mesh datasets, a synthetic dataset on SO(10) and alanine dipeptide with one restricted dihedral angle.

**Questions:**

Additionally to the questions in Strengths and Weaknesses:
- l.190 Why only "almost surely"?
- Could you apply your method to learned manifolds (e.g. as represented by an AutoEncoder)?
- Why does f(x, t) not appear in eq. (21, 22)?

**Ethical Concerns:**

["NO or VERY MINOR ethics concerns only"]

**Final Justification:**

The contributions of this work are twofold: It provides a theoretical characterization of the score function singularity when modelling on-manifold data with diffusion models and proposes two fixes.
While the authors acknowledge that this singularity has been known in the literature they claim to be the first to rigorously analyze its behavior in this way. While I am personally not very familiar with the related work on the topic, from the discussions of reviewers vZYH, TsMq and 42Gv it seems that the promised update to the related work section is sufficient to distinguish this work from previous ones on the topic.
Of the proposed methods only Niso-DM can consistently beat the best performance of the baseline method. Including the second method, Tango-DM, which performs worse due to the underperformant annealed sampler does not detract from the paper. There exist other methods, which incorporate the manifold structure that were not included as baselines in the paper. The authors have since provided these results in the discussion and will add them to the revised version of the paper. Technical questions of training on non-differentiable manifolds have been adequately explained in the discussion below.
I lean towards accepting the paper as it provides a rigorous theoretical analysis of the score singularity for manifold data and proposes a practical fix. My rating is not higher because, while the proposed fixes are more elegantly formulated than the naive solution of moving the data off-manifold before training, it is unclear whether they provide a consistent advantage in practice.

**Limitations:**

There are limitations related to runtime (sampling speed of the annealed SDE, manifolds with expensive projections) which, depending on how relevant they are in practice, should potentially appear in appendix F. I would be interested in how much slower the annealed SDE sampling is in the authors' experience. Otherwise the limitations seem to be complete.

**Quality:**

4

**Strengths And Weaknesses:**

**Strengths**
- The work gives a detailed characterization of the score function behavior for the original diffusion loss as well as the two proposed variants. Both are well motivated
- Experiments are extensive and oriented closely around benchmarks used for e.g. on-manifold ODE/SDE models.
- Niso-DM is a simple change to the noise schedule, which requires little design overhead and knowledge about the manifold to implement.
- Both training methods introduce little computational overhead, provided that projections to tangent/normal space are cheap
- Definitions and theoretical results are presented in a very comprehensible way, I especially enjoyed the introduction to diffusion models in Sec. 2.1

**Weaknesses**
- Projecting/calculating normal + tangential directions for noised samples in the embedding space can lead to problems if the manifold is not differentiable. Sec. 2.2 suggests that you assume a differentiable manifold, however this is contradictory to the experiments on mesh datasets. Can you explain how you compute normal/tangential components in regions of the embedding space that are projected to an edge or a corner of the mesh?
- Tango-DM can only be used in conjunction with the annealed sampler in the general setting. This sampler seems to not only under-perform in the experiments but is also computationally more expensive. Could you provide a rough overview of how much slower it is in practice?
- There are no baseline methods in the experiments, which incorporate any information about the manifold structure (e.g [1, 2]). Is there a reason you do not compare against them?

**Minor Weaknesses**
- For Niso-DM there exists a bias-variance trade-off between contracting closely to the manifold and avoiding the singularity. The authors analyze this trade-off in appendix E and show that it is still favorable over unmodified diffusion
- A naive solution to the singularity would be to just add (normal direction) noise to the training data (moving it off the manifold), then projecting the samples of the diffusion model back onto the manifold. This approach seems very closely related to your final version of Niso-DM, where the off-manifold noise level is set to a constant in the initial steps. Are there significant differences that I missed? Do you have a motivation for introducing the more general formulation of Niso-DM including $\alpha_t$ and have you investigated the effect of $\alpha_t$ apart from using it as in l.180?

In conclusion, the work provides insightful theoretical results and of the proposed novel methods, Niso-DM consistently performs well. The information on practical considerations and experiments seems to be only partially complete, which is why I will only vote to borderline accept for now.

[1] De Bortoli, Valentin, et al. "Riemannian score-based generative modelling." Advances in neural information processing systems 35 (2022): 2406-2422.
[2] Huang, Chin-Wei, et al. "Riemannian diffusion models." Advances in Neural Information Processing Systems 35 (2022): 2750-2761.

---

> ### Author Rebuttal · Authors · 2025-07-31
>
> We appreciate the reviewer’s valuable questions and suggestions.
>
> ## Strengths
> We appreciate the reviewer for their positive evaluation of our work. Our study provides a detailed characterization of the score function behavior for both the original diffusion loss and the two proposed variants, each of which is well motivated and thoughtfully developed.
>
> ## Weaknesses
>
> ### W1: Smoothness of Manifolds
>
> Indeed, when describing our algorithm, we assume the manifold to be smooth, and part of the mesh dataset satisfies this assumption. The manifold $\mathcal{M}$ corresponding to the mesh dataset is a polyhedron composed of triangles. On each face (triangle), the manifold is smooth, as it can be considered a part of a plane. **The non-smooth points are the shared edges between adjacent faces, which form a zero-measure set.** Therefore, this manifold is almost everywhere smooth, and our proposed algorithm is still applicable to this manifold without contradiction. In the implementation, for $x \in \mathcal{M}$, the normal direction of $x$ is the normal vector of the triangle (plane) that $x$ belongs to. The normal components are the projections of the score along this normal vector. For points lying on the shared edges (with zero probability of occurrence), we arbitrarily select the normal vector of one of the adjacent faces as the normal direction for that point. We will add more details about this example in the main text, as discussed above.
>
> ### W2: Performance of Tango
>
> We argue that Tango does not always underperform. In fact, our Tango algorithm still outperforms the vanilla diffusion model when using the annealed sampler. Moreover, in the dipeptide example, the optimal result was achieved with the Tango algorithm rather than Niso. At this stage, there is no definite conclusion regarding which method—Niso or Tango—is superior.
>
> While Tango tends to be slower due to the annealing sampler, **the runtime is primarily determined by the number of steps of the inner Langevin dynamics.** For example, in the toy experiment R2inR3, the sampling time for Reverse SDE is 1.68 seconds, while for Annealing SDE, the sampling time increases to 5.62 seconds with an inner step of 5 and to 9.40 seconds with an inner step of 10. The first phase of Annealing SDE (see Algorithm 3) requires only 0.82 seconds. Overall, **the computational cost of Annealing SDE is approximately 3–6 times higher than that of Reverse SDE.** These measurements, conducted on a standard laptop, highlight the relative computational expense. A detailed discussion on computational costs will be included in the appendix.
>
>
> ### W3: Baseline
>
> Following the reviewer’s suggestion, we have conducted the algorithm in [1] on our experiments for comparison. Diffusion models constrained to the manifold often rely on complex geometric information, such as the heat kernel or logarithmic mapping, to compute transition probabilities, which limits the computation of the denoising score matching loss. In contrast, the algorithm from Riemannian diffusion models (RDM, [2]), based on sliced score matching (or implicit score matching in [1]) and Hutchinson estimation, is applicable to more general manifolds. Therefore, we use the RDM algorithm as a baseline and conduct experiments on Mesh data and SO(10). However, RDM failed in the Alanine dipeptide case, which we suspect may be due to the limitation of sliced score matching when applied to SE(3)-equivariant distributions.
>
> Table 1 presents the comparison between the vanilla algorithm (with a final projection step), our method (using Niso+res with Reverse SDE), and RDM (baseline). Our method consistently outperforms the baseline. Specifically, in the SO(10) case, our approach outperforms RDM, while the vanilla algorithm does not. For Mesh data, RDM does not perform well, which can be explained by the fact that RDM relies on defining a diffusion process on the manifold to achieve a uniform distribution. Due to the geometric structure of the manifold, such as the narrow neck of "Spot the Cow", achieving a uniform distribution via SDEs on manifolds proves challenging.
>
> Table 1: Computational errors for Mesh data and SO(10). The results are reported as the mean and standard deviation over five independent runs.
>
> | Method   | Bunny           | Spot            | SO(10)          |
> |----------|------------------|-----------------|-----------------|
> | RDM      | 7.00e-1 (3.85e-3) | 7.22e-1 (3.32e-3) | 7.14e-3 (9.09e-4) |
> | Vanilla  | 3.52e-1 (2.17e-3) | 3.30e-1 (2.08e-3) | 9.49e-3 (1.91e-3) |
> | Ours     | **3.48e-1** (8.27e-4) | **3.28e-1** (6.19e-4) | **4.60e-3** (8.24e-4) |
>
>
> We appreciate the reviewer’s suggestion and hope that these additional experiments will be taken into account when reevaluating our work. We will include these additional experiments in the revised version of the paper.
>
>
> ## Minor Weaknesses
>
> ### W4: Bias-Variance Trade-off
>
> We thank the reviewer for carefully reading our appendix. As one of the fundamental philosophyies in machine learning, the bias-variance tradeoff phenomenon is widely observed in parameter selection scenarios. In this ablation study, we demonstrate that our algorithm is still favorable over unmodified diffusion. We do not consider this a weakness but rather evidence further supporting the advantages of our algorithm.
>
> ### W5: Non-isotropic Noise
> In the implementation, we chose $\sigma_t^{\alpha_t} = c_{\rm iso}$ as a constant, which aligns with the reviewer's understanding.  However, our approach is not entirely equivalent to pre-adding noise to the data before training, as our loss function (see Eq. (19)) employs non-isotropic score matching. The general formulation in the theorem was introduced to rigorously prove the scale differences from a mathematical perspective. While it is possible that selecting a more flexible $\alpha_t$ might lead to better results, for simplicity in our numerical experiments, we only investigated the case where $\alpha_t = \log c_{\rm iso} / \log \sigma_t$ in the paper.
>
>
> ## Questions
>
> ### Q1：'Almost surely'
>
> In the appendix, we prove that $\ell_{\text{tango}}(t,\theta) = \mathbb{E}_{x, x_t} \|s^{\parallel}_{\theta}(x_t,t)-P(x_t)\nabla_{x_t}\log p_{\sigma_t}(x_t)\|^2 + C_1$.
>
> Thus, when $s$, $p$, and $P$ are continuous functions, $s^{\parallel}_{\theta}(x_t,t) = P(x_t)\nabla_{x_t}\log p_{\sigma_t}(x_t)$ **always holds, not just "almost surely." We will remove the phrase "almost surely" in the updated version.**
>
>
> ### Q2: Learned Manifold Case
>
> In principle, our method can be extended to learned manifolds. As suggested by the reviewer, we use the AutoEncoder as an example. At first, we train an Encoder $\phi_{\theta_1}: \mathcal{M} \to \mathbb{R}^d$ and a Decoder $\psi_{\theta_2}: \mathbb{R}^d \to \mathcal{M}$, which provide a parameterized representation of the manifold. The subspace spanned by $\nabla\psi_{\theta_2}(\phi_{\theta_1}(x))$ corresponds to the tangent space $T\mathcal{M}_x$ at point $x$ on the manifold. Once the basis of the tangent space is obtained, the score function can be further decomposed into its tangential and normal components, which enables the implementation of our proposed methods. Intuitively, the success of our method on learned manifolds hinges on the Autoencoder's ability to accurately capture the underlying manifold structure. However, in this work, we do not plan to implement this approach and instead leave it as a direction for future research.
>
> ### Q3: f(x,t) Term
>
> In this paper, we focus on the Variance Exploding SDE (VESDE), characterized by $f(x, t) = 0$. To ensure clarity for readers, we will explicitly highlight this in the explanations of equations (21) and (22).
>
> ## Final Words
>
> Overall, our work provides a mathematically explicit analysis of the singular structure of the score function along its tangential and normal components, even though the divergence of the score function has already been widely studied. Furthermore, we propose two methods to improve the performance of diffusion models by mitigating the scale discrepancy of the score function. If one aims to improve accuracy by precisely computing the tangential component near $t=0$, we believe our approach captures the essence of the problem and has the potential to inspire future research.
>
> Once again, we sincerely thank the reviewer for their constructive feedback.

---

> > ### Comment · Reviewer_3q97 · 2025-08-01
> >
> > Thank you for your extensive response. I am, however, still unclear on some of the raised concerns.
> > **W1**: I agree that the non-smooth parts of the manifold have zero measure under the data distribution. However, even after the *first step* of the noising process a mix of tangential and normal noise (even if non-isotropic) can move data into regions, where it is projected to the non-smooth regions of the manifold. I would like to see additional considerations on how the solution of picking a normal direction at random from one of the adjacent faces affects the diffusion process and the correctness of the learned distribution.
> > **W2**: I agree that Tango DM seems to perform equivalently, if not better than the other methods when using the annealed sampler. My point was about the annealed sampler itself yielding worse results than the standard time reversal in all experiments, though I do concede that I missed the slightly improved result for alanine dipeptide. Overall, the requirement of the annealed sampler for Tango DM seems to be a significant downside. I still regard it as an interesting direction to explore, but am more convinced by the practical benefits of Niso DM.
> > **W3**: Thank you for including this additional baseline. It is especially interesting to see the (presumed) benefits of a noising process that can go off-manifold.
> > **W5**: Thank you for pointing out the differences between noising the data before training and constant $c_\text{niso}$ for Niso DM. It would be interesting to see how the two compare in practice.

---

> > > ### Author Response · Authors · 2025-08-05
> > > **About W1: Smoothness of the Mesh Manifold**
> > >
> > > We appreciate the reviewer’s prompt response. Below, we will further address some of the reviewer’s concerns.
> > >
> > > ### W1: Smoothness of the mesh manifold
> > >
> > > We first provide a clearer explanation of how the normal direction $n(x)$ is selected for $x \notin \mathcal{M}$.
> > >
> > > > For $x \notin \mathcal{M}$, the closest point $\pi(x)$ on a triangular mesh surface is determined as follows: First, $x$ is projected onto the planes of all triangular faces in the mesh using the face normals and vertex positions. Barycentric coordinates are then computed to check whether the projected points lie inside the triangles. If a point falls outside a triangle, it is further projected onto the nearest edge to ensure it remains on the triangle's boundary. Subsequently, the Euclidean distance between the original query point $x$ and all projected points is calculated, and the closest point $\pi(x)$ is selected by identifying the minimum distance.
> > > >
> > > >
> > > > Next, we assign the normal direction $n(\pi(x))$ of $\pi(x)$ as the normal direction for $x$ ($x\notin \mathcal{M}$). When $\pi(x)$ is located inside a triangle of the mesh, the normal of the triangle is chosen as the normal vector at $\pi(x)$ (i.e. $n(x)=n(\pi(x))$). **However, when $\pi(x)$ lies on an edge (or vertex) of the mesh, it is simultaneously associated with two (or three) triangles. In this case, we select the normal vector of the triangle with the smallest index as the normal direction for $\pi(x)$**. Here, we leverage the property of the *torch.argmin* function; see the *closest\_point* function in *manifolds/Mesh.py* in our supplementary materials.
> > >
> > > We realize that the use of the term "arbitrarily" in our previous reply was somewhat inappropriate. **Here, "arbitrarily" refers to the choice of $n(x)$ that does not follow a clear rule (In fact, we select the face with the smallest index according to the storage order.), rather than implying randomness.** Thus, for any $x$, the choice is deterministic and there is no randomness here. As a result, the normal vector $n(x)$ remains almost everywhere continuous, with discontinuities occurring only on a set of zero measure.
> > >
> > > However, we acknowledge that **at these discontinuous points, $n(x)$ (and consequently $P(x)$) undergoes sharp changes**, leading to $P(x)$ not being a continuous function in $\mathbb{R}^n$. We will include the detailed explanation above about the Mesh data experiment in the appendix.
> > >
> > >
> > > A possible solution to address discontinuities is to use neural networks to learn the level set representation of the manifold (See paper [*]), which aligns well with our setup. Specifically, the function $\xi$ is parameterized by a neural network and is trained such that $\xi(x)$ is close to zero and $|\nabla\xi(x)|$ is close to one, for $x$ on the mesh.
> > >
> > > The last three rows of Table 1 and Table 2 show the results obtained based on the learned manifold $\xi_{\theta}$. **We found that representing the mesh as a level set function results in worse performance. Therefore, although using the information from the mesh introduces discontinuities in $P(x)$, numerical performance ensures it an acceptable setup.** Theoretical analysis on the nonsmooth manifolds case needs further study in the future.
> > >
> > >
> > > - [*] Rozen, N., Grover, A., Nickel, M., and Lipman, Y. "Moser flow: Divergence-based generative modeling on manifolds." NIPS (2021).
> > >
> > >
> > > Table 1: Results for Bunny. First three rows: Directly using the mesh grid manifold (discontinuous), as described in the paper. Last three rows: Using neural networks to learn an approximate continuous representation of the manifold.
> > >
> > > |    | Method  | Reversal  | Annealing  |
> > > |:--------:|------|-----------|-----------|
> > > |  | Iso+res  | 3.52e-1 (2.17e-3)  | 3.94e-1 (1.63e-3)  |
> > > |  **Uncontinuous: From the paper**  | Niso+res  | **3.48e-1** (8.27e-4)  | 3.86e-1 (1.85e-3)  |
> > > |  | Tango+res  | -  | **3.84e-1** (1.57e-3)  |
> > > |  | Iso+res  | 3.79e-1 (8.26e-4)  | 4.26e-1 (2.12e-3)  |
> > > |  **Continuous: Based on $\xi_{\theta}$**  | Niso+res  | 3.77e-1 (6.18e-4)  | 4.31e-1 (1.56e-3)  |
> > > |  | Tango+res  | -  | 4.31e-1 (3.46e-3)  |
> > >
> > > Table 2: Results for Spot. First three rows: Directly using the mesh grid manifold (discontinuous), as described in the paper. Last three rows: Using neural networks to learn an approximate continuous representation of the manifold.
> > > |  | Method  | Reversal  | Annealing  |
> > > |:-------:|------|-----------|-----------|
> > > |  | Iso+res  | 3.30e-1 (2.08e-3)  | 3.48e-1 (1.82e-3)  |
> > > |  **Uncontinuous: From the paper**  | Niso+res  | **3.28e-1** (6.19e-4)  | 3.40e-1 (2.34e-3)  |
> > > |  | Tango+res  | -  | **3.39e-1** (1.74e-3)  |
> > > |  | Iso+res  | 3.84e-1 (1.53e-3)  | 3.97e-1 (3.20e-3)  |
> > > |  **Continuous: Based on $\xi_{\theta}$**  | Niso+res  | 3.82e-1 (1.67e-3)  | 3.98e-1 (2.09e-3)  |
> > > |  | Tango+res  | -  | 4.01e-1 (9.64e-4)  |

---

> > > ### Author Response · Authors · 2025-08-05
> > > **About W5: Non-isotropic Noise**
> > >
> > > ### W5: Non-isotropic Noise
> > >
> > > Taking SO(10) as an example, we compared the two methods under different constants $c_{\rm niso}$, with the results shown in Table 3. "Niso+res" refers to the method we used in the paper, while "pre-adding noise" denotes adding extra noise to the normal direction first, followed by using the isotropic denoising score matching loss.
> > >
> > > Currently, we cannot determine which of the two methods is better based on the results. However, we believe that our non-isotropic denoising loss, Eq. (19), provides a more elegant and unified formulation.
> > >
> > > Table 3: Comparason between the Niso+res and pre-noising algorithm under different $c_{\rm niso}$ with $\sigma_{\rm min}=0.001$.
> > > |  $c_{\rm niso}$  | 0.1  |  0.05  | 0.01  | 0.005  |  0.002  |
> > > |----------------|----------|----------|----------|----------|----------|
> > > | Niso+res  | 7.87e-3 (6.39e-4) | 7.92e-3 (2.27e-3) | 5.62e-3 (1.43e-3) | 5.47e-3 (1.60e-3) | 6.65e-3 (1.42e-3) |
> > > | pre-adding noise  | 8.05e-3 (1.01e-3) | 7.01e-3 (1.88e-3) | 7.33e-3 (1.56e-3) | 4.97e-3 (8.56e-4) | 6.95e-3 (1.64e-3) |
> > >
> > >
> > > **Finally, we would like to thank the reviewer for acknowledging the new baseline we provided.**

---

> > > > ### Comment · Reviewer_3q97 · 2025-08-06
> > > >
> > > > Thank you for these two follow-up experiments. I consider my concerns addressed and will keep my positive rating.

---

> > > > > ### Author Response · Authors · 2025-08-06
> > > > >
> > > > > We sincerely thank the reviewer for their valuable suggestions and the time and effort they dedicated to our work.
> > > > >
> > > > > Best regards,
> > > > >
> > > > > The Authors

---

### Official Review · Reviewer_42Gv · 2025-06-25

**Clarity:** 3
**Significance:** 2
**Originality:** 2
**Rating:** 4
**Confidence:** 4

**Summary:**

The paper studies the behavior of Variance Exploding Diffusion models when the data distribution is supported on a $d-$dimensional submanifold embedded in $R^n$. Based on this theoretical analysis, two new diffusion models are proposed, Niso-DM and Tango-DM, which can be used to learn densities supported on known low-dimensional sub-manifolds. Empirical studies are conducted to support the theoretical results.

**Questions:**

1. In section 2.1, what is the $\sigma_T^2$? You did not define it and the sentence in line 67 is confusing as $p_T$ approximates the true limiting distribution.

2. I don't understand Figure 1. The tangential component is in this case the 3. coordinate of the learned score function. Close to the manifold, that is $t\to 0$, this error decrease as seen by the funnel structure. However, the colorcode suggests that the average error increased closer to the manifold with the highest error for Iso+res. Would you mind elaborating?

3. I noticed different metrics for evaluating the performance on difference datasets. 1-Wasserstein, sliced Wasserstein, MMD, Jensen-Shannon divergence. Can you provide some reasons for that? Can you use a single metric for all? Can you provide a counterexample of a manifold where the assumption do not hold?

4. How strong are the assumptions of Theorem 3.1 and Theorem 4.1? In particular, the $M_1$ and $M_2$ boundedness?

**Ethical Concerns:**

["NO or VERY MINOR ethics concerns only"]

**Final Justification:**

The author resolved my 2 main concerns: first, the lack of discussion with [1] and other related methods, and second, the lack of comparison with another diffusion-based manifold method. I still think the method will benefit from using a proper score-function instead of an unconstrained one and encourage the authors to try it out.

Overall, the theoretical contribution is solid outweighting the limited evaluation with other methods. For the method to be seen as a high-impact in the area of learning on manifold densities for known manifolds, a more thorough evaluation is needed including Normalizing Flow and other methods. Therefore, I recommend a weak acceptance.

[1] *Density estimation on low-dimensional manifolds: an inflation-deflation approach*, Christian Horvat, Jean-Pascal Pfister.

**Limitations:**

The assumptions of Theorem 3.1 and Theorem 4.1 are not fully discussed, see Question 4 above.

**Paper Formatting Concerns:**

No issues.

**Quality:**

2

**Strengths And Weaknesses:**

Using the variance exploding SDE, the peturbed distribution at time $t$ is essentially a convolution of $p_0$ with the noise distribution $p_{t}(\tilde{x} | x)$. This observation is exploited in Theorem 3.1 to analyze the behavior of the true score close to the manifold and on the manifold when $t \to 0$. Using the resulting decomposition of the score, the diffusion loss functions can be neatly decomposed as well.

Theorem 4.1 provides a good theoretical analysis of Niso-DM showing that the score of $p_0$ on the manifold can be well approximated using the learned score under certain condition.

The proposed new algorithms are straightforward consequences of the theory and show good empirical results. Unfortunately, only the vanilla diffusion model is used as a benchmark. A comparison to other, similar methods is lacking. A practitioner who wants to decide which method to use for his/her manifold data is left behind.

In terms of novelty, a very similar analysis was conducted in the context of using Normalizing Flows to learn densities on low-dimensional manifolds [1]. In particular, Theorem 4 and Proposition 7 in [1] confirm equation (10), (12) and (18). Nevertheless, the decomposition (13) and the derivation of the Theorems is novel to the best of my knowledge. However, a thorough comparison to [1] is needed in the Related Work Section in my opinion.

Finally, I would like the authors to have a look at this paper [2]. They show that a conservative score function behaves very differently close to manifold data than a non-conservative score function. I wonder how your results will change when you use a conservative vector field as a score, that is, defining $s_{\theta}$ as the gradient of some scalar function. I conjecture that the results will improve even further.

Given insufficient related work section, the lack of comparison to other methods and some mathematical flaws (see additional comments below), I lean towards rejection in the current version.

[1] *Density estimation on low-dimensional manifolds: an inflation-deflation approach*, Christian Horvat, Jean-Pascal Pfister.
[2] *On gauge freedom, conservativity and intrinsic dimensionality estimation in diffusion models*, Christian Horvat, Jean-Pascal Pfister.


Some other interesting literature which might be interesting for you:
+ *Diffusion Models Encode the Intrinsic Dimension of Data Manifolds*, Jan Pawel Stanczuk


*Additional comments*:
+ line 38, mention that $d<n$.
+ mention citation [31] in Section 2.1
+ Please properly explain the $\nabla^{\mathcal{M}}$ operator in the main text.
+ Please use consistent notation for Expectation. Equation (4) is using $x \sim p$, equation (5) is using $x$ and Theorem 3.1 is using $p$.
+ Please explain all the relevant notation to understand Theorem 4.1. How is $\wedge$ defined?
+ Some References on volume forms on manifolds and equation (8) should be added in Section 2.2.
+ Typo in line 284. Table 1 is not in the Appendix.

---

> ### Author Rebuttal · Authors · 2025-07-31
>
> We appreciate the reviewer’s valuable feedback and suggestions.
>
> ## Strengths
>
> We thank the reviewer for their positive feedback on our work. In Theorem 3.1, we exploit the observation to analyze the behavior of the true score both close to and on the manifold, allowing for a neat decomposition of the diffusion loss functions. The proposed algorithms, derived directly from the theoretical insights, demonstrate strong empirical performance.
>
>
> ## Weaknesses
> ### W1: Baseline
>
>
> Following the reviewer’s suggestion, we have conducted Riemannian diffusion models (RDM, [*]) on our experiments for comparison. This algorithm, based on sliced score matching and Hutchinson estimation, is applicable to general known manifolds. Therefore, we use RDM as a baseline and conduct experiments on Mesh data and SO(10). RDM failed in the Alanine dipeptide case, which we suspect may be due to the limitation of sliced score matching when applied to SE(3)-equivariant distributions.
>
> Table 1 presents the comparison between the vanilla algorithm (with a final projection step), our method (using Niso+res with Reverse SDE), and RDM (baseline). Our method consistently outperforms the baseline. Specifically, in the SO(10) case, our approach outperforms RDM, while the vanilla algorithm does not. For Mesh data, RDM does not perform well, which can be explained by the fact that RDM relies on defining a diffusion process on the manifold to achieve a uniform distribution. Due to the geometric structure of the manifold, such as the narrow neck of "Spot the Cow", achieving a uniform distribution via SDEs on manifolds proves challenging.
>
> Table 1: Computational errors for Mesh data, and SO(10). The results are reported as the mean and standard deviation over five independent runs.
>
> | Method   | Bunny           | Spot            | SO(10)          |
> |----------|------------------|-----------------|-----------------|
> | RDM      | 7.00e-1 (3.85e-3) | 7.22e-1 (3.32e-3) | 7.14e-3 (9.09e-4) |
> | Vanilla  | 3.52e-1 (2.17e-3) | 3.30e-1 (2.08e-3) | 9.49e-3 (1.91e-3) |
> | Ours     | **3.48e-1** (8.27e-4) | **3.28e-1** (6.19e-4) | **4.60e-3** (8.24e-4) |
>
> We appreciate the reviewer’s suggestion and hope that these additional experiments will be taken into account when reevaluating our work. We will include these additional experiments in the revised version of the paper.
>
>
> [*] Huang, Chin-Wei, et al. "Riemannian diffusion models." Advances in Neural Information Processing Systems 35 (2022): 2750-2761.
>
> ### W2: Relation to Paper [1]
>
> We thank the reviewer for pointing out this highly relevant paper and its similarities to our work. Indeed, the idea of adding noise along the normal direction is similar between the two studies. We will include the following discussion in a revised version.
>
> > Paper [1] proposes adding noise along the normal direction to "inflate" manifolds, which is similar to our niso method. Theorem 4 and Proposition 7 in [1] analyze the relationship between the perturbed distribution and the original distribution on the manifold, which aligns with conclusions implied by our equations (12) and (18).
> >
> > On the other hand, there are several key differences. In [1], the added noise has a constant magnitude, requiring additional assumptions (e.g., Q-normally reachable) to prevent interference between noise at different points. In contrast, our analysis (Theorem 3.1) allows the additional noise magnitude to approach zero, relying on weaker assumptions. Additionally, [1] applies noise inflation prior to training the normalizing flow, whereas our method integrates noise addition directly into the diffusion process, aligning with the noise dynamics of diffusion models. Finally, while the analysis in [1] focuses on the perturbed density function, our work studies the asymptotic behavior of the score function.
>
> ### W3: Additional References
>
> We thank the reviewer for pointing out these related works. We will include citations to these papers in the Related Work section.
>
>
> ## Additional comments
>
> We thank the reviewer for their careful reading of our paper. We will make the necessary revisions accordingly. The operation $\wedge$ is defined as $a \wedge b = \min[a, b]$, and the definition of $\nabla^{\mathcal{M}}$ is as follows:  $\nabla^{\mathcal{M}}f(x) := P(x)\nabla_{x}f(x)$ for any function $f$.
>
> ## Questions
>
> ### Q1: Refrasing
> This section can indeed be written more clearly. In the revised version, we will rephrase the sentence as follows:
>
> > By carefully selecting $f$ and $g$, the distribution $p_{T}$ can be well approximated as a Gaussian distribution $\mathcal{N}(0, \sigma_T^2 I)$ at time $T$, where $\sigma_T^2$ is the variance of $X_T$.
>
>
> ### Q2: Explanation of Figure 1
>
> The tangential component corresponds to the first two coordinates (rather than the third) of the learned score function, as the manifold constraint ensures that the third component is zero. The tangential component is the first two coordinates of the learned score function. From a theoretical perspective, due to the scale discrepancy we mentioned, accurate estimation of the tangential component becomes harder as $t \to 0$, resulting in increased error. The experimental results are consistent with this theoretical analysis: the average error increased as $t \to 0$. Our method (the bottom two plots) reduces this error. The magnitude of the error is not related to the funnel structure. The funnel structure reflects the effective range of the distribution. As $t \to 0$, the distribution concentrates near $z = 0$, which leads to the funnel shape.
>
>
> ### Q3: Utilized Metrics
>
> In our setting, the data is strictly distributed on a submanifold of the space $\mathbb{R}^n$. **Ideally, metrics for evaluation should compare distributions based on distances defined on the manifold itself. However, such computations are generally infeasible. As a practical alternative, we calculate distances between distributions in the entire space** (with the exception of JS divergence for mesh data, which effectively captures differences on the manifold). Due to the unique properties of each manifold and the distributions defined on them, no single metric can reliably evaluate performance across all datasets. For instance, in the case of $\mathrm{SO}(10)$, which is embedded in a high-dimensional space, the 1-Wasserstein distance in $\mathbb{R}^n$ is computationally expensive and does not effectively capture distribution differences on the manifold, making sliced Wasserstein a better choice. Besides, for $\mathbb{R}^2$ embedded in $\mathbb{R}^3$, we observe that the variance of the 1-Wasserstein distance is large, while the variance of MMD is more stable.
>
> ### Q4: Assumptions in Theorems
> Our conclusions hold for points on the manifold, so in the proof, we only require local boundedness. Specifically, the following assumption is sufficient:
>
> $$
> \forall x \in \mathcal{M},\exists \delta > 0, \text{ s.t. } \sup_{y \in \mathcal{M}^{\delta}(x)} \max_{1 \leq i,j,j' \leq n} \left| \frac{\partial P_{ij}}{\partial y_{j'}}(y) \right| < +\infty,\text{ where }  \mathcal{M}^{\delta}(x)= [  \arg\min_{z \in \mathcal{M}} |\tilde{x} - z|^2 :  |\tilde{x} - x| < \delta ].
> $$
> To make the theorem statement more concise, we adopted a stronger assumption:
> $$
> \sup_{x \in \mathcal{M}} \max_{1 \leq i,j,j' \leq n} \left| \frac{\partial P_{ij}}{\partial x_{j'}}(x) \right| < +\infty.
> $$
>
> For compact manifolds, the uniform boundedness assumptions always hold, and consequently, the local boundedness assumptions hold. We can always assume that the manifold is bounded, as the sample data we obtain always have finite values.
>
> ## Limitations:
>
> We will include a discussion on the assumptions of Theorem 3.1 and Theorem 4.1 in the paper, as mentioned in the reply to Question 4 above.
>
>
> ## Final Words
>
> Overall, our work provides a mathematically explicit analysis of the singular structure of the score function along its tangential and normal components, even though the divergence of the score function has already been widely studied. Furthermore, we propose two methods to improve the performance of diffusion models by mitigating the scale discrepancy of the score function.
>
> While we acknowledge that some relevant papers were overlooked in the current version, we believe the novelty of our approach is still evident. **We will incorporate discussions of the related work in the revised version.** Once again, we sincerely thank the reviewer for their constructive feedback.

---

> ### Comment · Reviewer_42Gv · 2025-08-04
> **Response**
>
> Thank you for your response and careful consideration of my points. I appreciate the additional experiments and think they add value to the paper. I am also satisfied with the revised literature review. I am happy to adjust my score accordingly.

---

> > ### Author Response · Authors · 2025-08-05
> >
> > We sincerely thank the reviewer for their valuable suggestions and the time and effort they dedicated to our work.
> >
> > Best regards,
> >
> > The Authors

---

### Author Response · Authors · 2025-08-04
**Revised Related Work Section**

We once again thank the reviewers for their thoughtful feedback, as well as everyone involved for the time and effort they have dedicated to evaluating our work. In response to the reviewers' comments regarding the insufficient discussion of related works, we plan to add the following two paragraphs to the related work section: **Scale discrepancy of the score function** and **Additional noise along the normal direction**. If there are further suggestions, we remain open to improving this section.

> **Scale discrepancy of the score function.** The work [1] uses the singular value decomposition of the score matrix to identify the intrinsic dimension of the manifold, which essentially utilizes the discrepancy of the score function. References [2] [3] [4] [8] investigate the fundamental principles of diffusion models under the manifold hypothesis, by studying the eigen-decomposition of the Jacobian of the score functions. This analysis implicitly relates to the score discrepancy discussed in our work. In particular, the decomposition of the denoiser mapping (Eq. (12)) in [3], which involves a geometry-adaptive harmonic basis, also implies this discrepancy, aligning with Eq. (10) in our paper. Papers [5] and [6] study the Memorization and Generalization in Generative Diffusion under the manifold hypothesis through Hidden Manifold Models and Generalized Linear Models. Besides, [9] suggests that a conservative diffusion is guaranteed to yield the correct conclusions when analyzing local features of the data manifold.
>
> While previous works have implicitly touched upon the scale differences in manifold settings, our work explicitly formalizes this phenomenon with rigorous mathematical formulations and introduces novel solutions to address it. Furthermore, we believe our methods can enhance the accuracy of diffusion models during the manifold consolidation phase (as described in [2]) or the collapse transition times (as mentioned in [5] [6]), by refining the computation of the tangential component of the score function.
>
> **Additional noise along the normal direction.** Paper [7] proposes adding noise along the normal direction to "inflate" manifolds under the setting of normalizing flows, which is similar to our Niso method. Theorem 4 and Proposition 7 in [7] analyze the relationship between the perturbed distribution and the original distribution on the manifold, which aligns with the conclusions in our equations (12) and (18).
>
> On the other hand, our method differs from [7] in several key aspects. In [7], the added noise has a constant magnitude, requiring additional assumptions (e.g., Q-normally reachable) to prevent interference between noise at different points. In contrast, our analysis (Theorem 3.1) permits the noise magnitude to diminish toward zero, relying on weaker assumptions. Additionally, [7] applies noise inflation prior to training the normalizing flow, whereas our method integrates noise addition directly into the diffusion process, accompanied by the noise dynamics of diffusion models. Finally, while the analysis in [7] focuses on the perturbed density function, our work studies the asymptotic behavior of the score function.

- [1] Stanczuk, Jan, et al. "Your diffusion model secretly knows the dimension of the data manifold." arXiv preprint arXiv:2212.12611 (2022). Another name: Diffusion Models Encode the Intrinsic Dimension of Data Manifolds, ICML (2024).
- [2] Ventura, Enrico, et al. "Manifolds, Random Matrices and Spectral Gaps: The geometric phases of generative diffusion." ICLR (2025).
- [3] Kadkhodaie, Zahra, et al. "Generalization in diffusion models arises from geometry-adaptive harmonic representations." ICLR (2024).
- [4] Wang, Peng, et al. "Diffusion models learn low-dimensional distributions via subspace clustering." arXiv preprint arXiv:2409.02426 (2024).
- [5] Achilli, Beatrice, et al. "Memorization and Generalization in Generative Diffusion under the Manifold Hypothesis." arXiv preprint arXiv:2502.09578 (2025).
- [6] George, Anand Jerry, Rodrigo Veiga, and Nicolas Macris. "Analysis of Diffusion Models for Manifold Data." arXiv preprint arXiv:2502.04339 (2025).
- [7] Christian Horvat, Jean-Pascal Pfister. "Density estimation on low-dimensional manifolds: an inflation-deflation approach." JMLR (2023).
- [8] Wenliang, Li Kevin, and Ben Moran. "Score-based generative models learn manifold-like structures with constrained mixing." NeurIPS 2022 Workshop on Score-Based Methods.
- [9] Christian Horvat, Jean-Pascal Pfister. "On gauge freedom, conservativity and intrinsic dimensionality estimation in diffusion models." ICLR (2024).

---

### Decision · Program_Chairs · 2025-09-17

**Decision:**

Accept (poster)

**Comment:**

The paper studies the behavior of diffusion models under the realistic assumption of the manifold hypothesis, where the data distribution is supported on a lower dimensional manifold embedded in the ambient data space. The authors present theoretical analysis of the singularity that arises in the diffusion score function as it approaches the manifold in normal or tangential directions. Based on the analysis, two new diffusion models are proposed which try to resolve these singularities, and experiments are conducted to show improvements in the behaviour of these models.

The reviewers reached a consensus that the analysis in this paper is rigorous and useful. From the initial reviews, several issues were raised: failure to cite, discuss, and engage with the body of relevant literature; lack of baseline methods and ablations; unrealistic or impractical assumptions; and other more minor issues.

Most of the technical issues were addressed during the discussions, including with additional baselines and better discussion of limitations. The most common criticism was simply poor comparison to existing literature. The authors have given a longer write up of related work during the discussion which covered all the topics brought up by reviewers.

Hence, I am recommending to accept this submission. The authors must incorporate the changes brought up with reviewers into their final submission.